# Targeting BRF2 in Cancer Using Repurposed Drugs

**DOI:** 10.3390/cancers13153778

**Published:** 2021-07-27

**Authors:** Behnam Rashidieh, Maryam Molakarimi, Ammar Mohseni, Simon Manuel Tria, Hein Truong, Sriganesh Srihari, Rachael C. Adams, Mathew Jones, Pascal H. G. Duijf, Murugan Kalimutho, Kum Kum Khanna

**Affiliations:** 1QIMR Berghofer Medical Research Institute, Herston, QLD 4006, Australia; Simon.Tria@qimrberghofer.edu.au (S.M.T.); Hien.Truong@qimrberghofer.edu.au (H.T.); Sriganesh.Srihari@qimrberghofer.edu.au (S.S.); Rachael.Adams@qimrberghofer.edu.au (R.C.A.); Murugan.Kalimutho@qimrberghofer.edu.au (M.K.); 2Department of Biochemistry, Faculty of Biological Sciences, Tarbiat Modares University (TMU), Nasr Bridge, Tehran 14115-154, Iran; maryam.molakarimi@modares.ac.ir (M.M.); a.mohseni@modares.ac.ir (A.M.); 3School of Environment and Science, Griffith University, Nathan, QLD 4111, Australia; 4The University of Queensland Diamantina Institute, Faculty of Medicine, The University of Queensland, Brisbane, QLD 4102, Australia; mathew.jones@uq.edu.au; 5Institute of Health and Biomedical Innovation, School of Biomedical Sciences, Faculty of Health, Queensland University of Technology (QUT), Brisbane, QLD 4000, Australia; p.duijf@uq.edu.au; 6Centre for Data Science, Queensland University of Technology (QUT), Brisbane, QLD 4000, Australia

**Keywords:** BRF2, cancer, molecular dynamics simulation, drug repurposing, bexarotene

## Abstract

**Simple Summary:**

BRF2, a subunit of the RNA polymerase III transcription complex, is upregulated in a wide variety of cancers and is a potential therapeutic target; however, no effective drugs are available to target BRF2. The upregulation of BRF2 in cancer cells confers survival via the prevention of oxidative stress-induced apoptosis. In this manuscript, we report the identification of potential BRF2 inhibitors through in silico drug repurposing screening. We further characterized bexarotene as a hit compound for the development of selective BRF2 inhibitors and provide experimental validation to support the repurposing of this FDA-approved drug as an agent to reduce the cellular levels of ROS and consequent BRF2 expression in cancers with elevated levels of oxidative stress.

**Abstract:**

The overexpression of BRF2, a selective subunit of RNA polymerase III, has been shown to be crucial in the development of several types of cancers, including breast cancer and lung squamous cell carcinoma. Predominantly, BRF2 acts as a central redox-sensing transcription factor (TF) and is involved in rescuing oxidative stress (OS)-induced apoptosis. Here, we showed a novel link between BRF2 and the DNA damage response. Due to the lack of BRF2-specific inhibitors, through virtual screening and molecular dynamics simulation, we identified potential drug candidates that interfere with BRF2-TATA-binding Protein (TBP)-DNA complex interactions based on binding energy, intermolecular, and torsional energy parameters. We experimentally tested bexarotene as a potential BRF2 inhibitor. We found that bexarotene (Bex) treatment resulted in a dramatic decline in oxidative stress and Tert-butylhydroquinone (tBHQ)-induced levels of BRF2 and consequently led to a decrease in the cellular proliferation of cancer cells which may in part be due to the drug pretreatment-induced reduction of ROS generated by the oxidizing agent. Our data thus provide the first experimental evidence that BRF2 is a novel player in the DNA damage response pathway and that bexarotene can be used as a potential inhibitor to treat cancers with the specific elevation of oxidative stress.

## 1. Introduction

RNA polymerase III (Pol III) is critically important for the initiation of transcription. Transcription factor IIB-related factor 2 (BRF2) is a Pol III complex subunit which plays a key role in cellular function as a master regulator of oxidative stress, and its overexpression is linked to tumorigenesis [1]. Studies have shown high basal levels of BRF2 expression in breast and lung cancers [2,3], where it is a potential independent prognostic factor for the recurrence and metastasis of lung cancer. BRF2 was identified as a lineage-specific marker in lung cancer [4] and was later recognized as an oncogenic driver in breast cancer [5] and esophageal carcinoma [6]. The genetic activation of BRF2 has also been linked to lung squamous cell carcinoma and other cancers [7,8]. BRF2 knockdown can inhibit the migratory and invasive abilities of non-small cell lung cancer cells (NSCLCs) and induce the loss of the epithelial-mesenchymal transition [7,9,10]. Additionally, BRF2 has been shown to play a crucial role in breast cancers with heterogeneous *HER2* gene amplification. Within such tumors, only 10% to 15% of the cells show amplification of the *HER2* gene signature [3], whereas the remaining population is HER2-negative. By analyzing multiple TCGA cancer datasets, BRF2 has been shown to be significantly upregulated in an HER2-negative compartment of these tumors to compensate for the lack of HER2 amplification [3]. Multiple cancer models have also shown significant alterations in BRF2 copy number variations, specifically copy number gain and amplification [2]. Targeting specific Pol III components, such as BRF2 and other related molecular subunits, could be advantageous due to its specific role in regulating certain aspects of transcription and/or the oxidative stress response. We and others have shown a context-dependent role of BRF2 as a cis-associated ‘alternative driver oncogene’ with a significant role in lung and ER−/HER2+ breast cancers, respectively [2,4].

BRF2 recruits the Pol III complex to essential gene promoters, including 5s rRNA, transfer RNA (tRNA), and U6 small nuclear RNAs (snRNAs), hence influencing cell proliferation and growth [11,12]. All Pol III subunits function together with three transcription factors (TFs): TFIIIA, TFIIIB, and TFIIIC. Although TFIIIA is a single protein that is specifically recruited to type 1 promoters (5s rRNA), TFIIIB consists of three components: TBP (TATA binding protein), BDP1, and either of the TFIIB-related factors BRF1 (for tRNA) or BRF2 (only in type 3 or U6 promoter). The functions of TFIIIs are to recruit Pol III to the transcription start sites through a series of DNA–protein and protein–protein interactions. In order to target BRF2 effectively, the structure, interactions, functions, and regulatory role of this TF, among other subunits and components of the RNA polymerase III (Pol III) machinery, should be taken into consideration. BRF2 recruits RNA Pol III to type III gene external promoters, including the gene product selenocysteine (SeCys) tRNA, which is involved in the synthesis of selenoproteins, and acts to reduce reactive oxygen species (ROS) to maintain cellular redox homeostasis [1]. The absence or defective expression of selenoproteins induces apoptotic cell death [1]. Moreover, high baseline levels of BRF2 have been observed in cancer cell lines that are subjected to prolonged oxidative stress [1]. Therefore, it has been speculated that high levels of BRF2 expression in cancer cell lines support the sufficient expression of selenoproteins, in order to detoxify ROS and preserve redox homeostasis, which is a hallmark of cancer cells.

In the current study, we have identified a novel function for BRF2 in the regulation of the DNA damage response. Furthermore, we carried out in silico screening to target the BRF2-TBP-DNA complex and identified potential cancer drugs as candidates. This was followed by the experimental validation of the drug bexarotene to study its effect on BRF2 levels and its anti-proliferative effects.

## 2. Materials and Methods

### 2.1. TCGA Analyses for Genomic Instability

The correlations between gene expression (from The Cancer Genome Atlas (TCGA) RNAseq datasets) and 18 features of genomic instability (GI) are listed below:

(1) NtAI: The number of telomeric allelic imbalances [13]. (2) LST: large-scale transitions, the number of chromosomal breaks between adjacent regions of at least 10 Mb [14]. (3) HRD_LOH: homologous recombination deficiency score, the number of genomic segments with the loss of heterozygosity (LOH) [15]. (4) Sum_HRD_score: the sum of NtAI, LST, and HRD_LOH homologous recombination deficiency signature scores in (1–3) [16]. (5) wGII: weighted genomic instability index [17]. (6) ITH: intra-tumor heterogeneity, a measure for how heterogeneous an individual tumor sample is, determined using the ABSOLUTE algorithm [18]. (7) Ploidy distribution: the ploidy/DNA content; 2 is the equivalent of 2n, the normal/diploid amount, whereas 4 would occur after whole-genome doubling. The larger the deviation from 2, the more aneuploid the cells are. Ploidy was determined using the ASCAT (allele-specific copy number analysis of tumors) algorithm [19]. (8) Ploidy groups: in this panel, the ploidy data shown in panel (7) are binned, with “diploid” referring to tumor samples with 2n DNA content, “aneup_lo” referring to tumor samples that are aneuploid but near diploid/2n (either lower or slightly higher than 2n DNA content), “aneup_high” referring to tumor samples that are aneuploid and with a DNA content well above 2n, and “aneup_all” referring to all aneuploid tumor samples (the sum of the former two bins). (9) cal_scna_burden: the number of chromosome arm-level somatic copy number alterations. This is the total number of chromosome arms that are gained or lost, also known as chromosome arm aneuploidies (CAAs). This was determined using SNP6 array copy number data [20]. (10) wc_aneuploidy: the total number of whole chromosomes that are gained or lost. This was determined using SNP6 array copy number data [21]. (11) CIN70: a surrogate gene expression signature for chromosomal instability, based on 70 genes. The higher this score, the more chromosome instability is predicted [22]. (12) aneuploidy_score_AS: a score for aneuploidy, the sum of the number of chromosome arms and whole chromosomes gained or lost [23]. (13) genome_doublings: the number of whole-genome doublings, determined using the ABSOLUTE algorithm [18]. (14) silent_mutations_per_mb: the number of silent mutations per mega-base (MB) [24]. (15) nonsilent_mutations_per_mb: the number of non-silent mutations per mega-base (MB) [24]. (16) HRD_score_Peng: homologous recombination deficiency score. This is a gene expression signature, based on the expression of 230 genes. The higher the score, the more likely there are HR defects in the tumor [25]. (17) no_of_MSI_events: the number of microsatellite instability events per sample [26]. (18) chromothripsis: samples negative or positive for chromothripsis [27].

### 2.2. TCGA Analyses for Cellular Signaling and Immunoregulation

The Spearman correlation analysis results between the gene expression levels (mRNA levels from The Cancer Genome Atlas (TCGA) RNAseq datasets) and the variables are listed below:

(A) Scores for twelve pathways were determined via the integration of multi-platform data from several types of ‘omics’ analyses [28]. (B) The number of genes in the gene expression signature for the level of angiogenesis/ hypoxia indicated for each study 1. Angiogenesis score: 43 genes [29]. 2. Hypoxia score (Winter): 99 genes [30]. 3. Hypoxia score (West): 26 genes [31]. 4. Hypoxia score (Sorensen): 27 genes [32]. 5. Hypoxia score (Seigneuric): one gene [33]. 6. Hypoxia score (Ragnum): 32 genes [34]. 7. Hypoxia score (Hu): 13 genes [35]. (C) 1. Cancer/testis (CT) antigen load. 2. Indel neoantigen load. 3. SNV neoantigen load. (D) 1. Autophagy-related prognostic signature: a prognostic gene expression signature based on the weighted expression of eight autophagy-related genes [36]. 2. Stemness (mRNA): the level of stemness, based on mRNA data [37]. 3. Stemness (DNA methylation): the level of stemness, based on DNA methylation data [37]. 4. Predicted pCR to T/FAC chemotherapy: a 30-gene pharmacogenomic predictor of pathologic complete response (pCR) to preoperative weekly paclitaxel and fluorouracil-doxorubicin-cyclophosphamide (T/FAC) chemotherapy, determined based on 133 patients with stage I-III breast cancer [38]. 5. Proliferation score: a weighted proliferation gene expression signature [39].

For immunoregulation and immune cell infiltration, the levels of 26 tumor-infiltrated immune cell types, or states, were estimated using the tumor immune estimation resource [40].

### 2.3. Virtual Screening Procedure

A drug library, containing 8770 compounds in sdf format, was extracted from the DrugBank database v. 5.1.5 (https://www.drugbank.ca/) accessed on 24 May 2020 [41]. BRF2 and the structure files of drugs were converted into *pdbqt* format using AutoDock Tools4 software [42,43]. Peptide drug structures and ligand-containing Si atoms, due to the absence of force-field parameters, were excluded from the analysis. Eight thousand molecular dockings were performed, using the Autodock Vina 1.1.2 program [44]. Each drug was run through one docking simulation, which generated ten docking poses. A box of size x = 66.98 Å, y = 46.90 Å, z = 44.83 Å was placed over the full structure of the BRF2 structure. The ‘exhaustiveness’ value (the number of runs) was set at the default value of 8.

### 2.4. Molecular Dynamics Simulations for Protein–Ligand Complexes

The three-dimensional structure of BRF2 was extracted from the RCSB Protein Data Bank (PDB ID 4ROC), which is in complex with TBP and DNA molecules. Two sets of molecular dynamics (MD) simulations were performed. In the first MD simulation, we investigated the stable conformation of the ligand-free BRF2 structure before binding to TBP and DNA molecules. MD simulations and molecular mechanics (MM) minimization were performed using the Gromacs 2019.6 package under an amber ff99SB forcefield. MD simulations were carried out with periodic boundary conditions. Van der Waals forces and the particle mesh Ewald method were both used with a cut-off of 10 Å. The frequency used to update the neighbor list was 5. The protonation state of the Gromacs package was used to calculate the total charge of the protein. The protein was solvated with TIP3P water molecules with an 8-Å-radius buffer zone around the protein in a truncated octahedral periodic box. The systems were neutralized by adding the corresponding number of counteria (Na^+^ and CL^−^) using the genion module. MD simulation was performed in 4 steps. First, the whole system was minimized by the steepest descent, followed by conjugate gradient algorithms. Next, in the equilibration step, using a force constant of 1000 kJ/mol·nm, heavy atoms were restrained, allowing the solvent and ions to evolve by means of MD in the NVT ensemble (200 ps) and minimization. The equilibrium geometry was required to be at 298 K and 1 atm. To achieve this, we increased the system temperature and reassigned the velocities at each step based on the Maxwell–Boltzmann distribution equilibrated for 200 ps. We set the temperature coupling to 0.1 ps and the pressure coupling to 2 ps. For the thermostat and barostat during the equilibration, we used the V-rescale and Parrinello–Rahman algorithms, respectively, and all bonds were constrained using the LINCS algorithm. In the production step, which is the final step, we performed 100 ns MD simulations under an NPT ensemble. A Nosé–Hoover thermostat and a Parrinello–Rahman barostat were used, removing the position restraints to retain stable temperature and pressure in the production step. The temperature was at 310 K with a time step of 2 fs. Constraining the lengths of hydrogen-containing bonds was further improved through the addition of the LINCS algorithm.

Then, the output of the first MD simulation was used as the input for virtual screening. One of the best-ranked complexes from virtual screening was minimized and used for the second molecular dynamics simulations. To parametrize ligand molecules, a GAFF forcefield, which was assigned by the Antechamber program in AmberTools20, was used. The AM1-BCC charge model was used to calculate the atomic point. The coordinate file, in SYBYL mol2 format, was previously loaded to the Antechamber program in order to set these parameters. After generating the AMBER topology and coordinate files, these files were converted to GROMACS topology and coordinate files using the acpype conversion script. The other conditions were similar to those of the first MD simulation.

### 2.5. Binding Free Energy Calculations

The binding free energy calculations were performed as described previously [45,46,47]. During the equilibrium step, we analyzed the binding free energies of the complexes between BRF2 and ligands by taking 400 snapshots from the 60- to 100-ns MD simulations, using the g_mmpbsa tool in Gromacs [48].

### 2.6. Umbrella Sampling (US) Simulation

The US simulations were performed as described previously [48]. Briefly, we selected the average structure from the last 10 ns of the simulation, prepared using Gromacs 2019.6 software. We made the protein–ligand complex parallel to the y-axis and a box with a length of 12 nm was constructed to pull the ligand along the y-axis for a distance of 2.5 nm. Following neutralization, minimization, and equilibration, we performed US simulation using the center-of-mass-pulling method. Using a 600 kJ/(mol·nm) force, the ligand was pulled from the protein pocket over the course of 500 ps at the rate of 0.005 nm per ps. We saved the snapshots at each picosecond and 27 snapshots were extracted at intervals of approximately 0.1 nm, which were used as starting configurations for each US simulation (each of them independently simulated for 100 ps by performing NPT equilibration), following by performing 5 ns of mdrun. We calculated the potential mean force (PMF) from these US outputs using the weighted histogram analysis method (WHAM). The force (kcal/mol) needed to pull the ligand from the binding pocket and the corresponding distance pulled were demonstrated on the PMF graphs (y and x-axes, respectively). The binding free energy (ΔG) was calculated for this ligand by calculating the difference between the plateau region of the PMF curve and the energy minimum of the simulation.

### 2.7. Principal Component Analysis (PCA) and Free Energy Landscape (FEL)

PCA was performed, as described previously [48], to attain a mass-weighted covariance matrix of the protein atom displacement. This parameter is indicative of the dominant and collective modes of the protein from the total dynamics of the MD trajectory. A set of eigenvectors and eigenvalues reflecting the molecules’ concerted motion were extracted by diagonalizing the covariance matrix. To yield the eigenvalues and eigenvectors by calculating and diagonalizing the covariance matrix, we used g_covar, followed by the g_anaeig tool for plotting and analyzing eigenvectors, as well as determining the RMSF values for PC1 and 2. We used the gmx sham tool to calculate the free energy landscape based on the PC1 and PC2 eigenvectors. We performed free energy landscape (FEL) analysis to assess the conformational changes based on the PCA results. We were able to identify the stable or transient state of the biomolecules based on the free energy values of the conformations. The free energy, ∆G(X), was calculated using the Equation (∆G(X) = −KBTInP(X) [49]), where KB is the Boltzmann constant and T is absolute temperature, X represents the PCs, and P(X) is the probability distribution of the conformation ensemble along the PCs.

### 2.8. Cell Culture and Reagents

All cells were cultured based on ATCC instructions. Cultures were passaged every 4 days as per the manufacturer’s instructions (Stem Cell Technologies). HeLa iCas9 cells [50] were grown at 37 °C in Dulbecco’s modified Eagle’s medium (DMEM) with 10% tetracycline-free fetal bovine serum, 100 U/mL penicillin, and 100 U/mL streptomycin. All the cell lines were routinely tested for *Mycoplasma* infection by Scientific Services at QIMR Berghofer Medical Research Institute. Bexarotene was received as a gift from Hazel Quek, QIMR, Brisbane, Australia (Sigma Aldrich^®^, St Louis, MI, USA). All cells were treated according to the available IC50 value for bexarotene from Cancer Cell Line Encyclopedia (CCLE).

### 2.9. Gene Transduction and Transfection

For the transient BRF2 silencing (reverse transfection), we used 10–20 nM of small interfering RNAs (Shanghai Gene Pharma, Shanghai, China) and Lipofectamine RNAi MAX (ThermoFisher Scientific, Carlsbad, CA, USA) for 48 h. For the generation of stable and constitutive cell lines with knockdown of BRF2, we used small-hairpin RNAs (Sigma Aldrich^®^, St Louis, MI, USA). For transduction of the lentiviral construct, we used the spinfection method for 1 h in the presence of hexadimethrine bromide polybrene; (Sigma Aldrich^®^, St Louis, MI, USA). For the selection of clones, 5 µg/mL of Puromycin (Life Technologies^TM^) was used.

Sequence: CCGGCCTCAGGAAGTTAGGGACTTTCTCGAGAAAGTCCCTAACTTCCTGAGGTTTTT.

The lentiviral particles: SHCLNV-NM_018310 MISSION^®^, shRNA Lentiviral Transduction Particles.

### 2.10. CRISPR/Cas9-Mediated Genome Editing

Target sequences for BRF2 were selected from the Toronto KnockOut Library V3 (BRF2 gRNA1-GCAACTGCAGAACTCGACAA BRF2 gRNA2 CCAGTGGATATCCATCAGGG) [51]. Target sequences were ordered as 24-nt oligonucleotides with asymmetric 5′ overhangs, phosphorylated using T4 polynucleotide kinase, annealed, and cloned into BsmBI-treated lentiGuide-puro (Addgene 52963, Watertown, MA, USA) with T4 DNA ligase. For stable expression of gene-specific sgRNAs, lentiGuide-puro plasmids were cotransfected with psPAX2 and pMD2.G into Lenti-X 293T cells (Clontech, Palo Alto, CA, USA). Supernatants were filtered after 24 to 48 h, mixed 1:1 with fresh medium containing polybrene (10 μg/mL), and applied to target cells for 16 to 24 h. Transductants were selected in puromycin (5 to 20 μg/mL). Gene disruption was induced with doxycycline-regulated Cas9 transgene present in the host cell line [50].

### 2.11. Western Blot and Immunoblotting

Cells were exposed to genotoxic agents (ionizing radiation and cisplatin) and harvested at indicated timepoints. Protein lysate was made using Urea/SDS buffer lysis buffer (50 mM Tris pH 7.5, 7M urea, 1%SDS, 100 mM NaCl) and sonicated for 10–15 s. Protein quantification was carried out using a Pierce^TM^ BCA assay kit (Thermo Fisher Scientific, Waltham, MA, USA) as per the manufacturer’s protocol. For Western blotting, 30 µg of protein was loaded on SDS-PAGE gel and electrophoresed at 120 V. Proteins were transferred on an Amersham Hybond C nitrocellulose membrane, followed by staining in Ponceau S to confirm the transfer.

For immunodetection, the membrane was incubated for 1 h in blocking buffer, followed by overnight incubation (4 °C) with the indicated primary antibodies (listed in Table 1). After 3 washes with PBS/Tween and 1 h incubation with peroxidase-conjugated secondary antibody, followed by washing three times, the immunodetection step was undertaken. The detection of signals was based on the chemiluminescence reaction between peroxidase and a luminol/hydrogen peroxidase mixture from Western Lightning^®^ Plus ECL, followed by development using a ChemiDock Imaging System (BioRad, Hercules, CA, USA).

### 2.12. MTS Assay

The MTS cell viability assay (CellTiter 96^®^ Aqueous, Promega, WI, USA) was performed in a 96-well tissue-culture plate (Becton Dickinson Biosciences, Falcon^TM^Corning, Corning, NY, USA) with 1000 cells/well, seeded and treated the next day with drugs/siRNA for 72 h. One hundred microliters of complete DMEM media containing 10% MTS reagent was added to each well and incubated for 1 h and plates were read at 490 nm on a Biotek Powerwave™ XS2 microplate spectrophotometer (Winosski, VT, USA). Optical density (O.D.) was measured, recorded, and normalized to the respective control.

### 2.13. Proliferation Assay

Cells were plated in a 24-well plate with four replicates each, seeded at a density of 10,000 cells/well. The cells were allowed to incubate and image in the IncuCyte^®^ S3 Live-Cell Analysis system (Essen BioSciences Inc., Ann Arbor, MI, USA) for seven days at 3–4 h intervals. The data were analysed using the complementary Essen IncuCyte^®^ S3 Live-Cell Analysis software. Finally, the graphs were generated using the analysed data through GraphPad Prism 7.

### 2.14. Reactive Oxygen Species (ROS) Activity Assay

A Cellular ROS Assay Kit (ab113851) with a DCF-DA probe was used according to the manufacturer’s instructions to determine the ROS activity. The cell-permeant reagent 2′,7′-dichlorofluorescin diacetate (DCFDA, also known as DCFH, DCFH-DA, and H2DCFDA) quantitatively assesses ROS (hydroxyl, peroxyl and other radicals) in live cell samples. DCF is highly fluorescent and is detected via fluorescence spectroscopy with excitation/emission wavelengths of 485 nm/535 nm. For plate reader measurement, 10,000 cells were seeded in triplicate for each treatment condition in a 96-well plate. After 18 h of incubation post-seeding, bexarotene was added at a final concentration of 25 µM for 5 h. After washing with HBSS, cells were treated with DCF-DA (25 µM), incubated for 30 min. After washing, cells with or without bexarotene pretreatment were treated with either tBHQ (100 µM) or H_2_0_2_ (1 mM) for 30 min. The plates were then read at an extinction 485 nm and an emission wavelength of 530 nm.

### 2.15. Statistical Analyses

All statistical analyses were performed using GraphPad Prism v 8.0, using a general linear statistical model, as defined in each section. The error bar represents the mean ± standard error of the mean (SEM). The statistical significance of the *p*-value is designated with an asterisk (*); *p*-values: * *p* < 0.05, ** *p* < 0.01, *** *p* < 0.001, and **** *p* < 0.0001.

## 3. Results

### 3.1. Bioinformatics Analysis Identifies BRF2 as a Promising Target in Cancer Treatment

We identified BRF2 as a potential cis-associated driver gene in breast cancer by performing copy-number-altered network analysis [2]. Further examination of the literature revealed that aberrations in this region have been identified in breast tumors, correlating with metastatic relapse within one year [52,53]. To further validate this, we performed pan-cancer analysis on BRF2 expression levels and copy numbers using TCGA datasets. Comparison of RNA Seq data of gene expression levels in normal and tumor tissues in 32 cancer types reveals that BRF2 is significantly overexpressed in many tumor types, including malignancies of the breast, lung, liver, and kidney (Figure 1). In many cancers, such as esophageal squamous cell carcinoma, adrenocortical carcinoma, invasive breast carcinoma, NSCLC, bladder carcinoma, and sarcoma, *BRF2* is often amplified and/or gained through copy number alterations (Appendix A). The analysis of TCGA data on lung cancer revealed amplification/copy number gains in lung cancer, squamous cell carcinoma, and adenocarcinoma of 32%, 41%, and 23%, respectively (Appendix A). The verification of BRF2 against an independent dataset of 597 PAM50-subtyped breast tumors showed that BRF2 was upregulated, gained, or highly amplified in 37% of cases of TCGA, with copy-number amplification strongly correlating with mRNA upregulation in a substantial subset within each of the PAM50 subtypes (Appendix A) [54]. Furthermore, the METABRIC dataset showed a 14% amplification in breast cancer across 2509 patients (Appendix A). The analysis of 180 patients and 237 samples from the Metastatic Breast Cancer Project revealed that 32% of metastatic breast cancers exhibited amplification and upregulation of BRF2 (Appendix A). Correlation with survival data from these patients showed a significant association of BRF2 upregulation with poor survival, particularly in ER-positive tumors (Appendix A). Notably, we found that *BRF2* amplification was mutually exclusive to the loss of *BRCA1* and *BRCA2* in breast tumors, both in the TCGA and METABRIC datasets (Appendix A). The mutual exclusivity of BRF2 amplification and BRCA1/2 loss was also observed in breast cancer cell lines in the CCLE database (Appendix A). *BRCA1-* and *BRCA2*-associated tumors are defective in homology-directed DNA double-strand break (DSB) repair. *BRF2* amplification is rarely seen in the context of DNA damage repair deficiency. We have previously shown that BRF2 is upregulated in several breast cancer cell lines compared to near-normal mammary epithelial lines, and breast cancer cell lines with high BRF2 expression are dependent on it for cellular viability [2]. This analysis suggests that BRF2 is a potential therapeutic target in multiple cancers, including breast cancer.

Next, we analyzed the correlation between *BRF2* expression and 18 features (measures) of genomic instability, a hallmark of cancer. These analyses revealed that there were no strong correlations between the expression of *BRF2* in breast cancer datasets and the selected features of genomic instability (Appendix A).

We next investigated a possible link between BRF2 expression and cellular signaling pathways, as well as immunoregulation and tumor infiltration. The analysis of 12 different signal transduction pathways, focusing on breast cancer, identified a weak positive correlation between BRF2 expression and mTOR signaling (r = 0.1728, *p* = 1.9 × 10^−6^, *n* = 75); cell cycle regulation (r = 0.1133, *p* = 0.0018, *n* = 753); stemness (r = 0.1303, *p* = 1.9 × 10^−5^, *n* = 1072); proliferation score (r = 0.0856, *p* = 0.0051, *n* = 1072); and hypoxia score (r = 0.1513, *p* = 4.6 × 10^−7^, *n* = 1100) compared to other signaling pathways tested in breast cancer, including receptor tyrosine kinase (RTK), PI3K/Akt, Ras/MAPK, epithelial–mesenchymal transition (EMT), apoptosis, chemosensitivity, autophagy, neoantigens, cancer testis antigens, and DNA methylation (Appendix A). Finally, the levels of tumor-infiltrated immune cell types showed a marginal correlation between BRF2 expression and regulatory T cells (Tregs) (r = 0.0875, *p* = 0.0118, *n* = 828); CD8+ T-cells (r = 0.0715, p = 0.0189, *n* = 1078); T follicular helper cells (r = 0.0488, *p* = 0.1118, *n* = 1064), and activated NK cells (r = 0.0924, *p* = 0.0117, *n* = 744) in breast cancer, using a tumor immune cell estimation resource (Appendix A). In conclusion, there is a limited correlation between *BRF2* expression and cancer hallmarks, signaling pathways, and immune markers. However, its amplification is mutually exclusive to DNA repair defects in breast cancer, suggesting that overexpression of *BRF2* may result in DNA damage repair deficiencies.

### 3.2. Evaluation of the Role of BRF2 in the Regulation of the DNA Damage Response Pathway, Utilizing Normal Mammary Epithelial Cells and Breast Cancer Lines

Given the possible link between the overexpression of *BRF2* and DNA damage repair (DDR) deficiencies in tumors, as suggested by the bioinformatics data, we aimed to investigate the possible role of BRF2 in the regulation of DDR experimentally. Excess ROS causes severe damage to cellular macromolecules, especially proteins and DNA. ROS is known to activate DNA damage response signaling via the induction of DNA damage. BRF2 senses oxidative stress [1], and upon exposure to oxidative agents, BRF2-depleted cancer cells undergo oxidative stress-induced cell death, but normal cells avoid this through the upregulation of selenoproteins. Therefore, we suspected that a relationship exists between DDR signaling and BRF2. The exposure of MCF10A cells, a near-normal mammary epithelial cell line derived from human fibrosarcoma, to different doses of γ-irradiation showed a marked increase in BRF2 expression when harvested 1 h after exposure (Figure 2A). γH2AX, the biomarker for DSBs and p53-S15 and pKap1 (S824) phosphorylation, a downstream substrate of ATM/ATR, indicated the level of DNA damage induction in the cells. In addition, we detected a robust increase in BRF2 within 1 h of exposure to 0.5 Gy IR, which was more efficient than other markers of DDR signaling, including γH2AX and p-Kap1 (Figure 2A). Furthermore, a time-course experiment after exposing cells to 6 Gy IR revealed that BRF2 induction was very rapid and persisted over 6 h in MCF10A (Figure 2B). However, this increase was not mediated by an increase in the mRNA levels of BRF2 (Figure 2C). Similarly, in breast cancer lines, MDA-MB-231 and SUM159PT, using a longer time course of up to 24 h, we observed a similar rapid and persistent upregulation of BRF2 upon DNA damage (Figure 2D,E). Moreover, we observed a robust stabilization of BRF2, which correlated with an increase in γH2AX in MCF10A cells treated with tert-butylhydroquinone (tBHQ), the major metabolite of butylated hydroxyanisole, which induces an antioxidant response through the redox-sensitive nuclear factor-E2-related factor-2 (NRF2) transcription factor (Figure 2F). NRF2 is a transcription factor that regulates the expression of antioxidant proteins and protects against oxidative damage [55]. This experiment further links BRF2 to DDR through the induction of oxidative stress. Additionally, we used the chemotherapeutic drug cisplatin as another DNA-damage-inducing treatment; the concentration of drug used and length of exposure did not generate reactive oxygen species (data not shown), which ionizing radiation is known to cause. BRF2 is upregulated following cisplatin treatment (Figure 2G), which implies a link between DNA damage induction and BRF2 upregulation. Notably, after cisplatin treatment, we did not observe changes in the level of the oxidative stress marker NRF2 (on the contrary, a slight decrease was evident at 6 h) under the same experimental conditions. This supports a link between BRF2 upregulation and DNA damage induction, although we cannot entirely rule out the possibility that BRF2 upregulation is triggered by cisplatin-induced oxidative stress.

Furthermore, we found that knocking down BRF2 using siRNA (MCF10A cells) caused an increase in baseline and IR-induced DNA damage, which was assessed via immunoblotting for γH2AX; however, no difference was observed in the IR-induced activation of pKAP1 (S824), a direct substrate of ATM signaling (Figure 2H). Likewise, the γH2AX clearance kinetics after IR-exposure were also markedly delayed when MCF10A cells were stably BRF2-depleted using shRNAs (Figure 2I,J). Collectively, these data suggest a potential role for BRF2 in the regulation of DNA damage repair, given that BRF2-depleted cells showed persistent DNA damage.

### 3.3. Targeting BRF2 to Interrupt Its DNA Binding

Given the upregulation of BRF2 in cancer and our findings indicating its link with DNA damage repair, we aimed to target BRF2 by repurposing available drugs. BRF2 interacts with TBP and this complex can bind to DNA [1] (Figure 3A). The structure of BRF2 consists of a Zn ribbon, N/C cyclin repeats, and an arch, which binds the C-terminal and TBP anchor domains to the rest of the protein (Figure 3B,C). Initially, we attempted to target different available pockets of this protein in order to inhibit its functions, particularly Cysteine 361 (C361), which is considered to be an oxidative stress switch in BRF2 [1]. Appendix A lists the drugs from the DrugBank and NCI databases which showed the best energy binding affinity, and Appendix A shows the architecture of these drugs at the C361 binding site, according to the molecular docking mechanism. However, the molecular dynamics (MD) simulation for all of these drugs in the pocket, as well as the binding site to DNA, failed to demonstrate a stable bond (data not shown).

Given the difficulty involved in targeting BRF2 at the C361 site, we aimed alternatively to inhibit the BRF2–TBP binding complex. For this purpose, we extracted the BRF2 structure from PDB using 4ROC pdb ID, this structure contains BRF2 in complex with TBP (Figure 3D). Then, we used MD simulation to investigate the stable form of BRF2 in a free state. A 100-ns (100,000 ps) MD simulation was performed to investigate the dynamic stability of free BRF2. The root-mean-square deviation (RMSD) was calculated to compare the initial and final structural conformations of the protein backbone. During the course of the simulation, the relative stability of the protein to its initial conformation was determined using the RMSD value, wherein a smaller value was indicative of greater stability. In order to evaluate the stability of both systems, the RMSD value for the Cα backbone was calculated over 100 ns. The RMSD profile indicated that, during the initial periods of simulations, the BRF2-free structure deviated considerably from the X-ray structure in the complex form [1]. The RMSD curves showed that the backbone trajectories of the free-BRF2 structure were stable and reached equilibrium after the first 30,000 ps of the simulation, which ranged between 8 and 9Å (Figure 3E). Furthermore, our 3D-structural analysis showed that at the end of the MD simulation, domain 2 (BRF2-CTD) of the protein was rotated clockwise (Figure 3F, Appendix A). This conformational change in BRF2 after binding TBP is an important feature that might play a key role in its binding to DNA. Therefore, we aimed to investigate this change of conformation in more detail.

### 3.4. Principle Component Analysis (PCA) and Free-Energy Landscape

For further evaluation of the BRF2 conformational changes after binding to TBP during the MD simulation, principle component analysis (PCA) was carried out [56,57]. Figure 4A represents a plot of the eigenvalues obtained from the diagonalization of the covariance matrix of the Cα atomic fluctuations. The reduction in the eigenvalue amplitude indicates a shift from concerted motions to more constrained, localized fluctuations. This analysis suggested that the first two eigenvectors obtained from PCA can account for a higher percentage of the total motions in all simulations. As seen in Figure 4A, the properties of motions for BRF2 described by the first two PCs were different.

To better understand the conformational changes of the BRF2 protein in the free state, the conformational spaces of BRF2 were generated to gain significant information, using projections of MD trajectories on the first two PCs. Furthermore, to deeply understand the motions, the displacements of eigenvectors 1 and 2 were calculated (Figure 4B). As shown in Figure 4B, the overall motion of the BRF2 protein has a different subspace of structures during 100-ns MD simulations, especially in PC1 mode. Based on Figure 4B, the protein visits three conformational clusters in PC1. The root-mean-square fluctuation (RMSF) analysis of PC1 showed that these clusters correspond to enhanced displacements in the regions of 1–30, 70–95, and 115–130 residues (Figure 4C).

To study the differences in the motion behaviors in the PC1 and PC2 modes, two porcupine plots are displayed in Figure 4D which employ the first and second eigenvector using VMD software. The direction of the arrow is an indicator of the collective motion direction, and the length of the arrow expresses the strength of the movements. Based on this Figure, the conformation of free BRF2 is different from that of the TBP-DNA-BRF2 complex form of this protein.

In order to determine the low-energy basins (minima) and the stability of the protein explored during the simulation, the free energy landscape (FEL) values of BRF2 were constructed using the projections of their own first (PC1) and second (PC2) eigenvectors (Figure 4E). The energy minima and energetically favored protein conformations are shown by dark blue spots, and the unfavorable conformations are indicated by yellow spots. The shallow and narrow energy basin observed during the simulation showed the low stability of BRF2. Based on this figure, BRF2 showed a cluster consisting of two connected energy minima basins close to each other.

### 3.5. Targeting BRF2–TBP Binding Using Drugbank Ligands

After identifying the conformational changes of BRF2, we used the correct conformation for in silico studies. To interpose the BRF2-TBP binding complex using an effective drug from the DrugBank database, we performed a virtual screening process using molecular docking with the rigid protein and flexible ligands. The optimized output structure of BRF2 after MD was utilized as a target for the virtual screening phase, and the top drugs were ranked based on the lowest binding energy value. The list of drugs (Table 2) and the architecture of binding are presented in Figure 5A–H and Appendix A. As shown in Table 2, the docking results showed that the binding energy of phthalocyanine to BRF2 was higher than that of the other ligands (−10.5 kCal/mol). Therefore, the stability of this ligand was analyzed during the MD simulation. The conditions of our MD simulation were explained in the Material and Methods section. As shown in Appendix A (0 ns) and Appendix A (after 8 ns) the ligand separated from the binding site and moved out of the cavity (the direction of the ligand’s movement is indicated by arrow). Nevertheless, phthalocyanine, the large aromatic macrocyclic compound, had the lowest energy binding; this compound was not stable after simulation. As can be seen in the docking representations (Figure 5), the second hit was the retinoic acid receptor gamma agonist (CD564), which can fit properly at the pocket of the site.

Therefore, we chose CD564 for simulation and performed MD for both BRF2 and the drug (Figure 6A, Appendix A). The RMSD value of the ligand was stable up to 16,000 ps, followed by a sharp rise from 0.6 to 0.9 Å, with fluctuations around this value until the end of the simulation. The analysis shown in Figure 6A indicated that the RMSD value of the protein reached equilibrium and oscillated around 0.2 nm for approximately 7000 ps of the simulation time. This evidence clearly indicates that the whole system was stable and equilibrated and that CD564 is a suitable compound to inhibit the binding site of BRF2 and TBP. The final structure of the complex and ligand binding site is shown in Figure 6B–D and Appendix A.

### 3.6. Thermodynamic Parameter Calculations

One of the most important analyses after MD is the calculation of binding free energy (ΔG). Using g_mmpbsa (molecular mechanic/Poisson–Boltzmann surface area, MMPSA) software and molecular dynamic simulations, we calculated the relative binding free energy. Four hundred extracted snapshots were used to calculate the binding energy. The binding free energy and its related components, acquired from the MM/PBSA calculation of the ligand-BRF2 complexes, are listed in Table 3. The results showed that the ligand possessed a high negative binding free energy value of −412.059 kJ/mol. Further evaluation showed that electrostatic energy contributed the most to the protein binding to the ligand, where the carboxyl group bound to Arg81 with electrostatic interactions, causing the total electrostatic energy to become negative (Figure 6E). Our analysis showed that the binding energy of this residue was −24.29 kJ/mol. Furthermore, the binding energy for other residues is available in Appendix A. Van der Waals interactions also played a critical role in this binding, as the ∆G_solv-polar_ value was 197.664 KJ/Mol. The ∆G_MMPBSA_ value was also indicative of strong, high-affinity binding between the protein and ligand.

To calculate the absolute ligand-protein complex binding energy, the US method was used and the ligand was pulled 2.5 nm from the binding pocket. The binding free energy (ΔG) was calculated by measuring the difference between the highest and lowest value of the PMF graphs. This unbinding process requires −10.29 kcal/mol energy to dissociate the ligand. As shown in Figure 6E, the ligand has carboxyl and aromatic ring sub-structures. Based on this LigPlot, the ligand has strong interactions with surrounding residues, especially TRP131, PHE180, GLN84 and ARG81. The pulling simulation revealed that an extended stay inside the binding pocket was the result of interactions with several amino acids. Figure 6F and Appendix A represent the unbinding pathway of the ligand from the binding site. During the dissociation process, we observed four energy minima (Figure 6G). The conformations of the four mentioned energy minima are shown in Figure 6G. At conformation shown in Figure 6(H1a), the carboxyl group of the ligand made a hydrogen bond and exhibited electrostatic interactions with Tyr260 and Arg81, respectively. Furthermore, rings 3 and 4 established pi-pi stacking and pi-slkyl interaction with Trp131 and Tyr176, respectively. At the second energy minimum Figure 6(H1b), the mentioned hydrogen and electrostatic interactions were broken, but the pi-pi and pi-alkyl interactions remained. At the third energy minimum Figure 6(H1c), the carboxyl group of the ligand exhibited a new electrostatic interaction with Lys181. At the fourth energy minimum Figure 6(H1d), the length of the salt-bridge interaction between Lys181 and the ligand reached 2.71 to 4.58 angstroms, so the ligand started to come out from the binding cavity. The molecule again pulled continuously, and the PMF graph demonstrated equilibration at a distance of around ~3 nm. At this position, the molecule was entirely unbound and became solvent-exposed.

### 3.7. Experimental Validation of the Drug

In order to experimentally validate the drug’s efficacy, we searched for an available FDA-approved drug that was the most similar to CD564. According to our DrugBank search, bexarotene (Figure 7A) had more than 70% structural similarity, as well as a similar function as a retinoid that activates retinoid X receptors (RXRs) [58]. Bexarotene (Targetin) is an approved anti-cancer drug for the treatment of cutaneous T cell lymphoma (CTCL) [59]. To further examine whether this drug is as efficient as CD564, the geometry of bexarotene in the binding site was initially studied via MD. Based on the docking results, the binding energy of the ligand at the binding site is equal to −8.8 kCal/mol. As shown in Figure 7B, the geometry of the ligand at the binding site is similar to that of CD564.

The stability of the BRF2-bexarotene (Bex) complex was analyzed during a 100-ns MD simulation. The RMSD plot of this simulation, shown in Figure 7C, indicates that the RMSD value of the protein increased gently up to 30 ns of simulation. Then, the RMSD value suddenly increased to 0.5 nm after 36 ns, which is indicative of significant conformational changes in the complex (Figure 7C). Then, the RMSD value decreased again and reached approximately 0.3 nm at the end of the simulation. Based on Figure 7C, there was minimal variation in the RMSD of the ligand during the MD simulation. As shown in Figure 7C, significant conformational changes were observed in the protein. For the purpose of better visualization, we colored residue numbers 83 and 84 at two conformations (before MD (green) and after 37 ns (cyan)) (Figure 7D). Based on these observations, after 37 ns, the left domain of protein rotated upward. Furthermore, a similar situation in the ligand can be seen. However, at the end of the simulation, the conformation of the protein had returned to its normal form (Figure 7E). These data showed that Bex was a potential candidate for experimental validation.

After initial in silico confirmation, we aimed to identify the drug sensitivity response based on BRF2 gene expression using the genomics of drug sensitivity in cancer (GDSC) database [60]. The IC50 of Bex correlated negatively to the expression of BRF2 among the breast cancer cell lines (the threshold of expression set to −2 to 2) (Figure 7F), unlike the other available drugs on the list (alectinib and AZD5991) (Appendix A). This is indicative of the potential efficacy of Bex, with increasing BRF2 expression in the indicated cancer setting.

To experimentally validate the effect of Bex on the BRF2-TBP binding complex, we treated an MCF7 control cell line (scrambled siRNA) and a BRF2-depleted cell line (BRF2 specific siRNA) with either DMSO or the drug. BRF2-depleted cells showed a decline in cell viability (MTS assay) compared to control MCF7 cells, which is indicative of the dependency of the cells on this essential gene. However, a further reduction of cell viability was observed in both Bex-treated scramble and BRF2-knockdown cells, suggesting that the anti-proliferative effect of Bex observed in this line is non-selective for BRF2 expression (Figure 7G). Western blotting of drug-treated MCF7 cells did not show a significant decrease in BRF2 expression compared to untreated cells. Consistently with this, we failed to observe a significant induction of cell death, as evident by the cleaved PARP in the Bex-treated MCF7 cells (Figure 7H), which was clearly evident after siRNA-mediated depletion of BRF2.

We also examined the doxycycline-induced CRISPR/iCas9-mediated knockdown of BRF2 in HeLa cells. We did not observe significant changes in the BRF2 expression level after Bex treatment in doxycycline-negative cells used as a control and doxycycline-induced cells (knockdown of BRF2 HeLa cells). However, Bex treatment of the control (Dox−) and BRF2-knockdown cells (DOX+) resulted in a dramatic decline in the oxidative stress agent tBHQ-induced expression of BRF2 (Figure 7I). Consistently with this, the proliferation assay showed a dramatic reduction in cell proliferation in both the control (Dox−) and BRF2-knockdown cells treated with both tBHQ and Bex (Figure 7J). Next, we aimed to assess whether this effect was due to the role of bexarotene in quenching ROS generation by tBHQ. Therefore, we treated the cells with tBHQ with or without pretreatment with bexarotene. The results showed that the pre-treatment of cells with bexarotene significantly reduced ROS formation with and without tBHQ treatment (Figure 7K).

## 4. Discussion

It has been proposed that targeting transcription factors may offer a therapeutic option with lower toxicity and increased efficacy than current chemotherapy drugs, which creates opportunities for novel drug-development approaches [61]. Classical chemotherapy agents targeting DNA integrity and cell division have the potential to give rise to secondary malignancies. The current strategies target oncogenic molecules or signaling pathways that tumor cells are dependent on for growth and survival. This phenomenon is known as oncogene addiction. However, there is emerging interest in targeting multi-component cellular machinery that is not directly involved in growth and proliferation to maintain the malignant properties that contribute to tumorigenicity (non-oncogenic addiction) [62,63]. BRF2 and the other class III genes transcribed by Pol III are known as ‘housekeeping genes’, and their expression is required in all cell types for cellular viability. Housekeeping genes have been suggested as anti-cancer targets with therapeutic potential given the likelihood of reduced associated toxicity [64]. TFs are targeted by approximately 10% of all FDA-approved anti-cancer drugs [65]. However, most TFs are not targetable because of their localization, or because their components and subunits do not possess enzymatic activity, which is usually easier to target with specific drugs. Therefore, we endeavored to identify a mechanism by which BRF2 could be targeted. However, most of the accessible pockets and available domains in BRF2 were not targetable, and we could only target BRF2 and TBP binding sites to impair their DNA-binding capacity. The formation of a complex between various proteins has a key role in many biological processes. Consequently, preventing complex formation interrupts the associated processes. However, BRF2 plays a key role in protecting cancer cells against oxidative stress and our data additionally suggest that BRF2 plays a role in DDR regulation. Therefore, the inhibition of this target could be more valuable than targeting other TFs or housekeeping genes.

Recently, there has been increased interest in targeting polymerases, such as the targeted destruction of Pol I, to treat patients with advanced solid tumors, which ultimately expands therapeutic opportunities [66]. Notably, targeting components of Pol I using the selective inhibitor CX5461 to treat tumors with MYC amplification has shown the selective killing of tumor cells through the activation of non-canonical ATM signaling [67]. Additionally, different modes of action for CX5461, such as TOPO isomerase II poison or G-quadruplex-mediated global DNA damage induction, have been proposed [68,69]. Regardless of the difficulties in targeting RNA polymerases and their associated regulators, the recent discoveries of drugs that are able to target these proteins underline their potential in the future of cancer therapy. Oncogenes such as MYC, or tumor-suppressors such as RB and p53, are multiple regulators of Pol III and Pol II. Furthermore, these common regulators have a global effect on Pol III, which makes it more difficult to assign a specific gene contribution to Pol III deregulation. Due to its pivotal role in cell growth, proliferation, and tumorigenesis, Pol III activity is tightly regulated by a series of activators and repressors. For example, the Pol III repressor MAF1 is negatively regulated by a series of phosphorylation events on seven different serine residues [70]. Similarly, RB and the RB-related family members repress Pol III activity and heightened levels of Pol III transcripts are implicated in accelerated growth associated with RB dysfunctional tumors [71,72]. Casein kinase 2 (CK2) is an extremely conserved kinase that plays a pivotal role in many aspects of cell physiology; moreover, CK2 is often overexpressed in several human cancers [73]. Among its many substrates, CK2 also regulates several transcription factors and polymerases, including Pol III [74] and the TFIIIB sub-factor TBP [75]. Another kinase involved in Pol III phosphorylation is ERK. In response to mitogen activation, the mitogen-activated protein (MAP) kinase ERK regulates Pol III activity through direct binding and the phosphorylation of BRF1 [76]. The phosphatase PTEN plays a fundamental role in tumor suppression by inhibiting PI3K signaling, and it is often mutated or lost in human cancers [77]. Moreover, PTEN represses the activity of Pol III by blocking the formation of functional TFIIIB complexes. Specifically, PTEN blocks the association between BRF1 and TBP; this inhibition occurs through the reduction of phosphorylation of BRF1, which induces the dissociation of TFIIIB [78].

Considering the vital role of TFs and their subunits in regulating oncogenic pathways, we performed a comprehensive bioinformatic analysis, correlation studies, and a literature review on the effect of BRF2 on signaling pathways. However, our analysis revealed that unlike the Pol III transcript itself, the BRF2 transcript is not linked to the above-mentioned signaling pathways. Notably, BRF2 mRNA expression in human tumors also did not correlate with any feature of genomic instability as an indicator of DNA damage (Appendix A). This is supported by our in vitro data in human breast cancer lines (Figure 2), showing that BRF2 protein levels are regulated after the exposure of cells to DNA-damaging agents, rather than at the transcript level. Thus, it is likely that BRF2 might be post-transcriptionally regulated by the above-mentioned signaling pathways.

Next, we further characterized the relationship between BRF2 and DDR experimentally. BRF2 is a major oxidative stress regulator, and oxidative stress mechanisms can drive tumorigenesis [2]. A BRF2-regulated gene product, known as SeCys tRNA, is involved in the synthesis of selenoproteins. The main function of SeCys tRNA is to reduce reactive oxygen species (ROS) levels and maintain cellular redox homeostasis through selenoprotein synthesis [1]. The absence or defective expression of selenoproteins induces apoptotic cell death. Studies have shown a high basal level of BRF2 expression in breast [2] and lung cancer [3], as well as in cancer cell lines such as HCC95 which are subjected to prolonged oxidative stress [4]. Therefore, it has been speculated that the high levels of BRF2 expression in cancer cell lines assist in maintaining the sufficient expression of selenoproteins in order to detoxify ROS and preserve redox homeostasis. DNA damage induction is linked to oxidative stress, which can in turn cause further DNA lesions. It was conspicuous that in the investigated cancer cell lines and near-normal breast cell line (MCF10A), the expression of BRF2 was upregulated after ionizing radiation. We also used cisplatin as another DNA damage-inducing treatment, as published reports from many laboratories have demonstrated that DNA is a direct target of cisplatin toxicity [79,80,81]. The most revealing evidence was the hypersensitivity to cisplatin seen in DNA-repair-defective prokaryotic and eukaryotic cells. However, there are some reports that the formation of reactive oxygen species depends on the concentration of cisplatin used and the length of exposure. In our in vitro experiments, cells were only transiently treated with cisplatin for up to 6 h, and we did not detect any generation of reactive oxygen species after cisplatin exposure in our experimental condition, assessed using the DCFDA/H2DCFDA ROS Assay kit (data not shown). Thus, the cisplatin experiment represents DNA-damage-inducing treatment to exclude BRF2 as possibly being activated by oxidative stress, which is triggered by radiation [81,82,83,84]. BRF2 demonstrates clear upregulation following cisplatin treatment, which implies a possible link between BRF2 and the induction of DNA damage. Additionally, the expression of NRF2 (a protein stimulated by oxidative stress [85]), was examined after cisplatin treatment and under our experimental conditions, no significant changes were detected. This strengthened the possibility of the link between BRF2 and DNA damage. While we were preparing the revised manuscript, the link between Pol III and DDR was reported, in which Pol III localizes to the site of double-strand breaks (DSB) and is required for DNA repair [85]. Consistently with our findings, the authors reported an increase in the chromatin levels of the Pol III-subunits in cells treated with DSB-inducing agents [85].

After characterizing the potential role of BRF2 in DDR, we initially aimed to target C361 in BRF2, which is critical on/off switch of ROS. However, according to our molecular simulations, we failed to identify any effective drugs. Therefore, we concentrated on targeting the BRF2-TBP-DNA complex. However, after MD simulation, the high protein RMSD value along the trajectory indicated that a significant change in the molecular structure had occurred. The stability of the protein after 30 ns, persisting at a value of approximately 1 nm, illustrated its stability after the large conformational displacement. Following this discovery, we used PCA to reveal the critical motions of the protein, as a confirmation of conformational changes. The PCA revealed the origin of the large conformational difference between the initial and final MD structures. This was supported by their different directions of motion relative to each other. Our analysis of the residue displacements, according to the projection of the MD trajectories on the first and second components, verified the larger displacements of some residues, located at the N-terminus of the protein, which exhibited two conformations, visiting different subspaces during the MD simulations. This provided significant dynamics information involving the conformational diversity of the BRF2 protein in both the bound and free states. The free energy landscape (FEL) analysis provided intimate details about the stable conformation and energy minima of the protein. BRF2 FEL analysis showed two energy minima that represent the two conformations of the protein (bound and unbound). Finally, the stable conformation of this step was used as a receptor structure to perform virtual screening.

A structure-based virtual screening method was used to select the ligand with a high binding energy value using the stable conformation of BRF2 before binding to TBP. Our analysis showed that the ligand has an acceptable binding energy value. The MD simulation method was used to study the stability of the ligand at the binding site. An RMSD plot confirmed that the protein and the ligand were stable during the MD simulation. In the next step, the relative binding free energy of the ligand to the protein was calculated using the MMPBSA method. MMPBSA analysis showed that the non-polar solvation energy, and the Van der Waals and electrostatic interactions (the most effective) contributed to the total interaction energy negatively. However, only the polar solvation energy contributed to the total free binding energy positively. Furthermore, the interaction plot generated by ligplot showed that the carboxyl group at the ligand exhibits electrostatic interaction with Arg81 of the protein. Collectively, using in silico approaches, we identified putative inhibitors of the BRF2-TBP-DNA complex after the virtual screening of more than 3500 FDA-approved drugs or drugs under investigation in clinical trials in the DrugBank database. We identified CD564 as a promising candidate that could create a stable binding interaction with the binding site of BRF2-TBP and consequently interfere with the functionality of BRF2. We validated these results using an MD simulation, in which the RMSD results indicated the stability of the bound and thermodynamic parameters, verified by the binding energy. We finally confirmed the binding of CD564 to BRF2 using the molecular dynamics simulation.

Due to the limited availability of CD564, we queried similar FDA-approved drugs to examine the results. The FDA-approved drug bexarotene (Bex) exhibited the best score (70% similarity) and therefore it was chosen for primary in silico analysis. Because it was successful in binding and molecular simulation through docking, as determined by MD, we further validated the effectiveness of this drug experimentally. First, we found a negative correlation between the drug sensitivity and the expression of BRF2 in breast cancer cell lines, which indicates that this drug is more efficient when BRF2 expression is higher. However, Bex treatment did not impact BRF2 expression and slightly reduced the proliferation of control and BRF2-depleted MCF7 cells without the induction of cell death, suggesting that the anti-proliferative effect of Bex was independent of BRF2 expression. However, siRNA-based depletion of BRF2 in MCF7 led to a significant induction of cell death. Similarly, Bex treatment alone was unable to reduce BRF2 expression significantly in both Dox negative (control) and positive (BRF2-knockout) cells, generated using the CRISPR/iCas9 system. This was in line with our in silico data, which showed that the drug binds to BRF2 in a certain conformation. Surprisingly, tBHQ was able to increase the BRF2 level approximately twofold in both conditions. This phenomenon indicates that the CRISPR/iCas9 system could not totally knock out BRF2 expression and upon treatment with tBHQ, which generates oxidative stress, its expression increased. Due to polyclonal selection of this line, heterogeneous populations of cells might still express BRF2 to some extent even in Dox-positive lines, which are indispensable for cell viability. Interestingly, Bex significantly reduced tBHQ-induced BRF2 upregulation in both control and BRF2-knockdown cells. Additionally, a combined Bex and tBHQ challenge in these lines significantly reduced cell proliferation but treatment with Bex or tBHQ alone did not. We hypothesized that the drug was only effective on a particular conformation of BRF2 before its binding to TBP. This finding is in line with our hypothesis that the drug is not effective on the changed BRF2 conformation in complex with TBP, which exists in the cells. However, the newly synthesized BRF2, following the induction of oxidative stress in both conditions, was inhibited by the drug. This scenario can confirm the anti-cancer effects of this drug in cancers with high oxidative stress that upregulate BRF2 expression. However, further experimental investigations are required in order to test the binding capacity of the purified recombinant BRF2 protein to the drug, which is outside the scope of this paper.

However, this drug can also modulate different cellular components and signaling pathways. Bexarotene is a retinoid X receptor (RXR) agonist, which is FDA approved for the treatment of cutaneous T cell lymphoma. As a therapeutic strategy, RXR agonists have been investigated in other cancers. Specifically, bexarotene downregulates the expression of inflammatory cytokines, such as interleukin (IL) 6, IL-8, and monocyte chemoattractant protein-1 [86] Additionally, in TNF-α stimulated human fibroblast-like synoviocytes, bexarotene has been shown to downregulate the expression of matrix metallopeptidases (MMPs), upregulate the anti-inflammatory cytokines IL-4 and TGF-β1, and attenuate nuclear factor (NF)-κB and p38 mitogen-activated protein kinase (MAPK) signaling [86]. Given the role of BRF2 in reducing reactive oxygen species (ROS), we examined the specific role of the drug on the generation of ROS by the oxidizing agent tBHQ and the related ROS activity. We found that in our experimental conditions, Bex pretreatment could significantly reduce baseline as well as oxidant-tBHQ-induced ROS levels. The anti-proliferative effect of the combination of Bex and tBHQ can be explained through a significant reduction in levels of BRF2. Our results suggest the potential efficacy of Bex in reducing BRF2 levels in cells with a high level of oxidative stress, but future validations are required in order to investigate the detailed mechanism of the drug’s action in vitro and in vivo.

## 5. Conclusions

In conclusion, we showed BRF2 to be a potential therapeutic target in cancer by comprehensively analyzing the publicly available pan-cancer data. We also discovered the regulation of BRF2 after DNA damage in breast cancer. Notably, using virtual screening on more than 3500 available drugs, we found inhibitors of the binding of the BRF2-TBP-DNA complex, which were successfully validated by molecular docking, energy binding, and MD simulation results. The MD results revealed that BRF2 undergoes a conformational change after binding to TBP, which leads to changes in the available pockets. Using this structure, we found that bexarotene, an FDA-approved drug, is a potential hit for the development of BRF2-specific inhibitors, according to in silico analysis and the negative correlation of drug sensitivity to BRF2 protein expression in the GDSC database. Further experiments showed that this drug inhibited the proliferation of control and BRF2-depleted cells, suggesting that its anti-proliferative effects are not selective for BRF2 expression. Interestingly, bexarotene reduced the oxidative-stress-induced high level of BRF2 by reducing the cellular levels of generated ROS. Overall, this study identified BRF2 as a promising cancer target; indicated the molecular changes in the architecture of this protein via binding to TBP; provided the potential hit compounds for this target, followed by examining one of them experimentally, as well as examining new roles for BRF2 in DDR regulation.

## Figures and Tables

**Figure 1 cancers-13-03778-f001:**
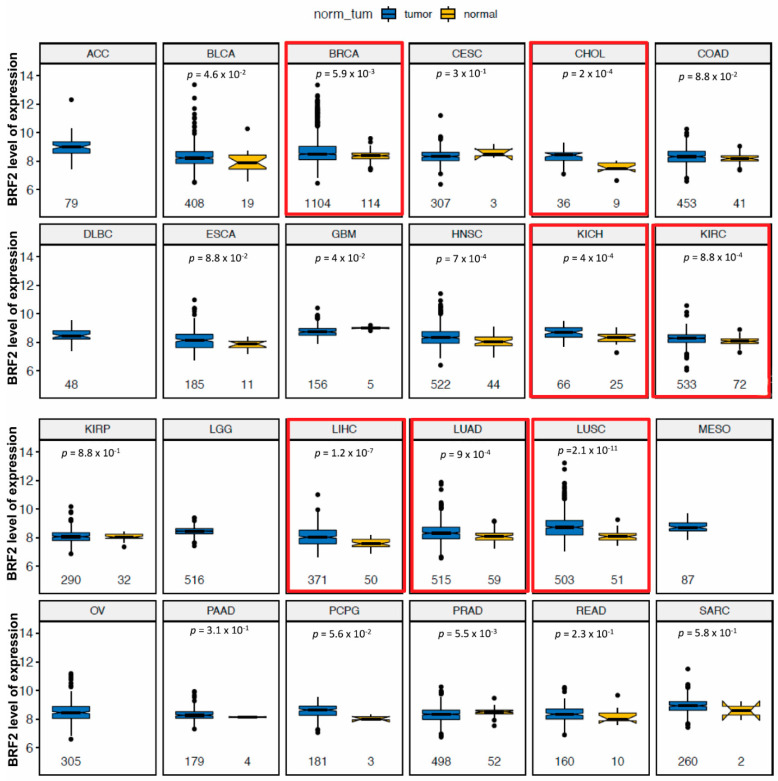
BRF2 elevated expression in multiple tumors. Comparison of gene expression levels in normal and tumor tissues in 24 cancer types. All data in these box plots are from The Cancer Genome Atlas (TCGA) RNAseq datasets. The box plots compare mRNA expression levels for the indicated gene (*y*-axes) between tumors and matched adjacent normal tissues (*x*-axes) of 24 cancer types. The abbreviations in the grey bars on top refer to the cancer types. Sample sizes (n) are shown below each box. Normal samples are missing for a few cancer types. The *p*-values were calculated using the (unpaired) Mann–Whitney U test. (For cancer type abbreviations, see the Appendix A).

**Figure 2 cancers-13-03778-f002:**
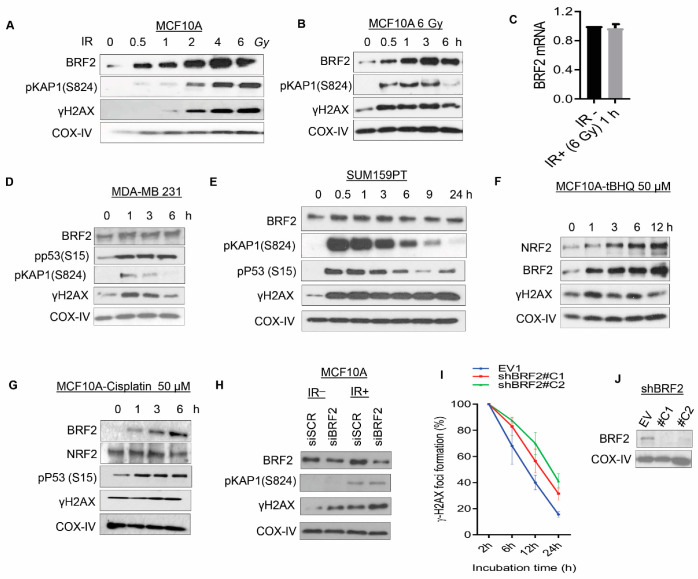
BRF2 is a potential DNA damage regulatory gene. (**A**) The dose-response of BRF2 and other DNA damage response (DDR)-related proteins in MCF10A after exposure to the indicated dose of ionizing radiation (IR). (**B**) Time course of BRF2 expression following 6 Gy IR in MCF10A. (**C**) BRF2 mRNA expression in MCF10A post-1 h exposure to IR (6 Gy). (**D**,**E**) Time course of BRF2 expression following 6 Gy IR in breast cancer cell lines: (**D**) MDA-MB-231 and (**E**) SUM159PT. (**F**,**G**) Expression of BRF2, NRF2, and other DDR-related proteins in MCF10A cells treated either with (**F**) 50 µM tBHQ or (**G**) 50 µM cisplatin at indicated time points. (**H**) Effect of BRF2 knockdown using siRNA on DDR signaling proteins (pKAP1(S824) and γH2AX) in MCF10A cells with and without IR exposure (6 Gy, post-6 h). (**I**) γH2AX foci clearance assays were evaluated after exposure to 2 Gy IR in MCF10A cells over the time course of 24 h. (**J**) Immunoblotting shows the expression of BRF2 in empty vector (EV) and two different BRF2 knockdown clones (C1, C2) established in MCF10A; COX IV served as a loading control.

**Figure 3 cancers-13-03778-f003:**
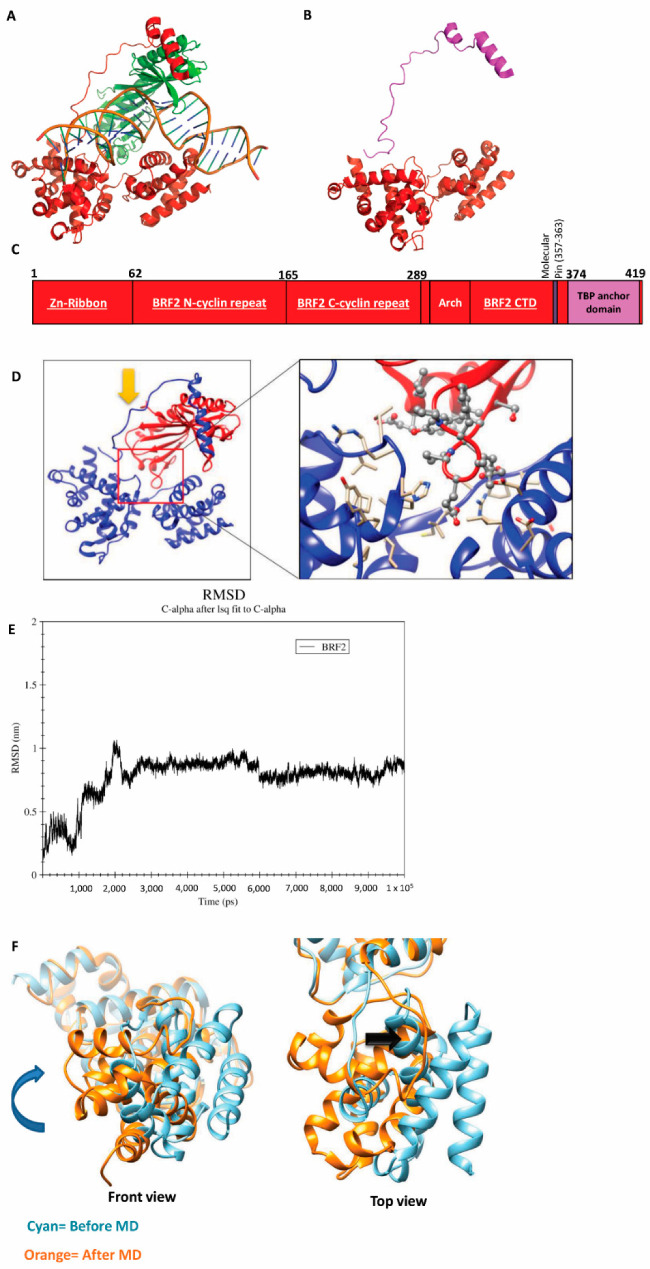
BRF2 structure and conformational changes. (**A**)The structure of BRF2 in complex with TBP and DNA. (PDBID: 4ROC). (**B**) The structure of BRF2. (PDBID: 4ROC). (**C**) Schematic representation of BRF2 domains. (**D**) The binding site of TBP on the surface of BRF2 (PDBID: 4ROC). (**E**) RMSD plot of free-BRF2 during the 100 ns MD simulation by means of GROMACS. The Figure depicts a representative image of at least three independent repeats. (**F**) Structural changes of BRF2 at the end of the MD simulation. Cyan: before MD, orange: after MD (see more details in Appendix A).

**Figure 4 cancers-13-03778-f004:**
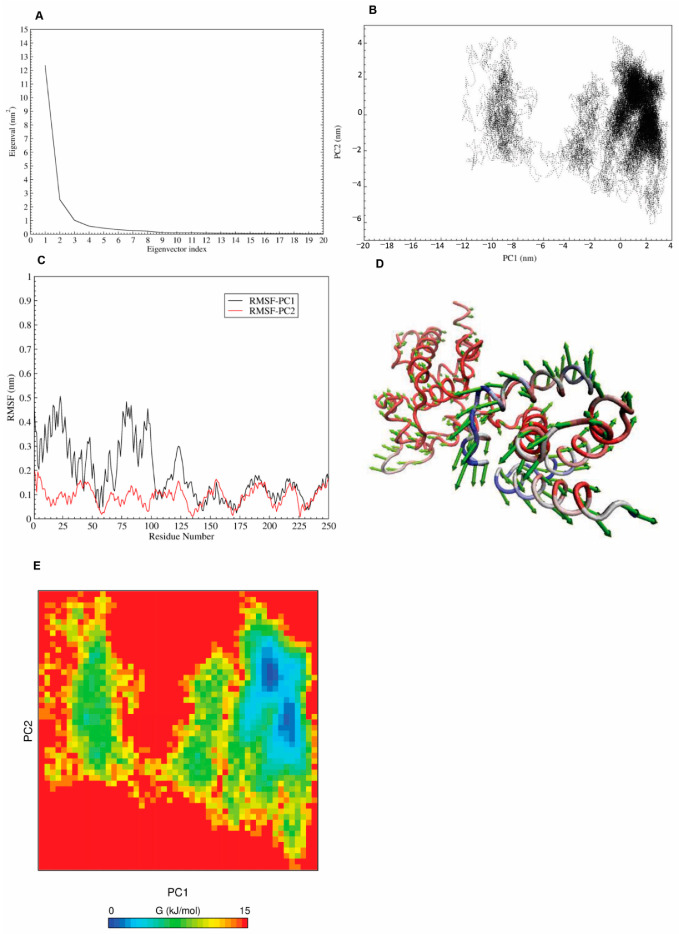
Confirmation of BRF2′s conformational changes using the PCA method. (**A**) The plot of eigenvalues obtained from the Cα covariance matrix built from MD trajectories. (**B**) Principal component analysis of the BRF2 protein. (**C**) Residue-based mobility plot of BRF2 in modes 1 and 2. (**D**) Porcupine plots of the PCA analysis for PC1, depicting the movement and altitude of the C-alpha atoms throughout the 100 ns of simulation. (**E**) Free energy landscape of BRF2 protein.

**Figure 5 cancers-13-03778-f005:**
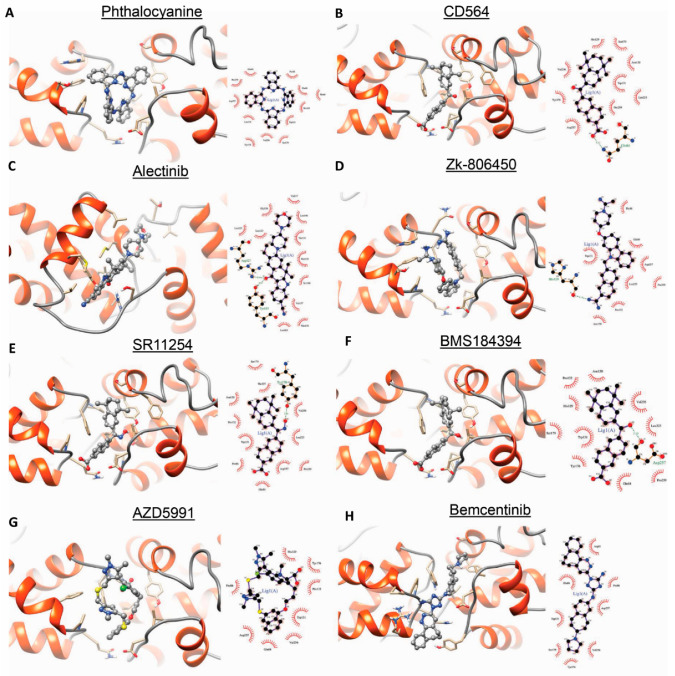
Results of the 8 top-ranking docked compounds with the BRF2 protein according to their binding energies. (**A**,**B**) surface views of the two best-ranked ligands according to docking. (**C**–**H**) protein–ligand 3D and 2D interaction diagrams of the protein ligand complex.

**Figure 6 cancers-13-03778-f006:**
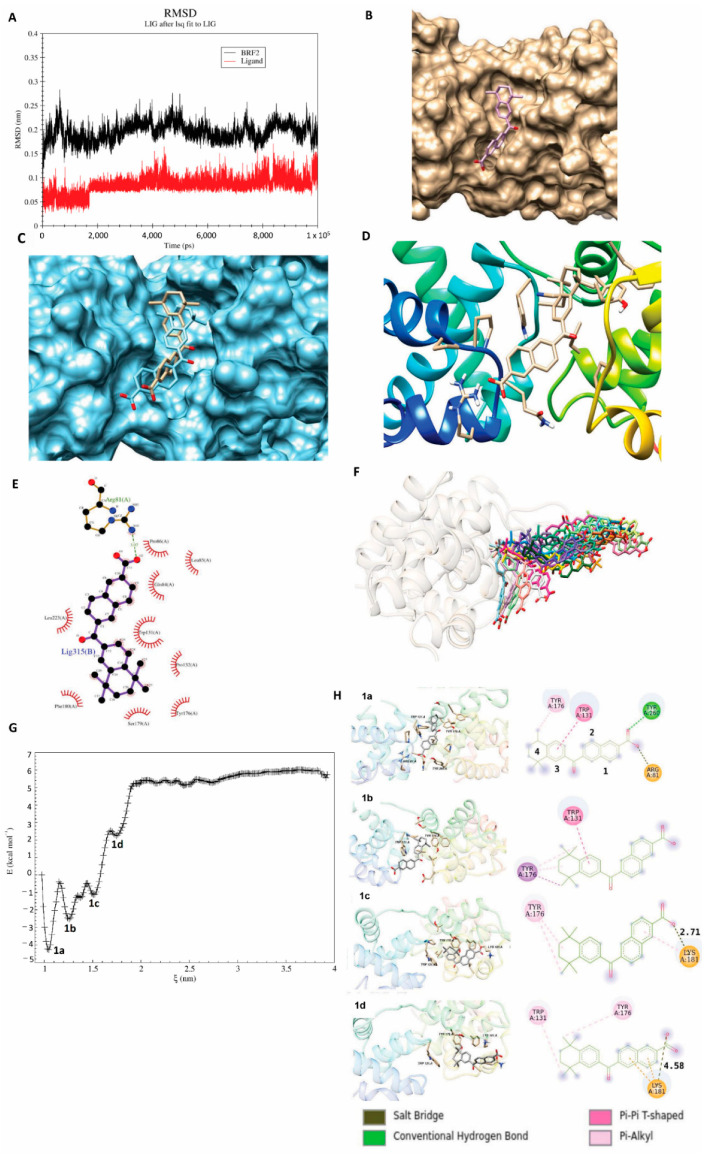
Molecular dynamic simulation. (**A**) RMSD of BRF2 (black) and the ligand (red) during the 100-ns MD simulation. (**B**–**D**) Docked BRF2-ligand complex presented in different views. (**B**) Surface view, (**C**), surface view showing the conformational changes of the ligand before (khaki) and after (blue) MD in the pocket, (**D**) ribbon view. (**E**) LigPlot-generated snapshot of the ligand and residues in the active site. (**F**) the unbinding pathway of the ligand from the binding site. (**G**) The PMF graph obtained from US simulation, (**H**) 1a, 1b, 1c, and 1d are the energy minima obtained through the US simulation.

**Figure 7 cancers-13-03778-f007:**
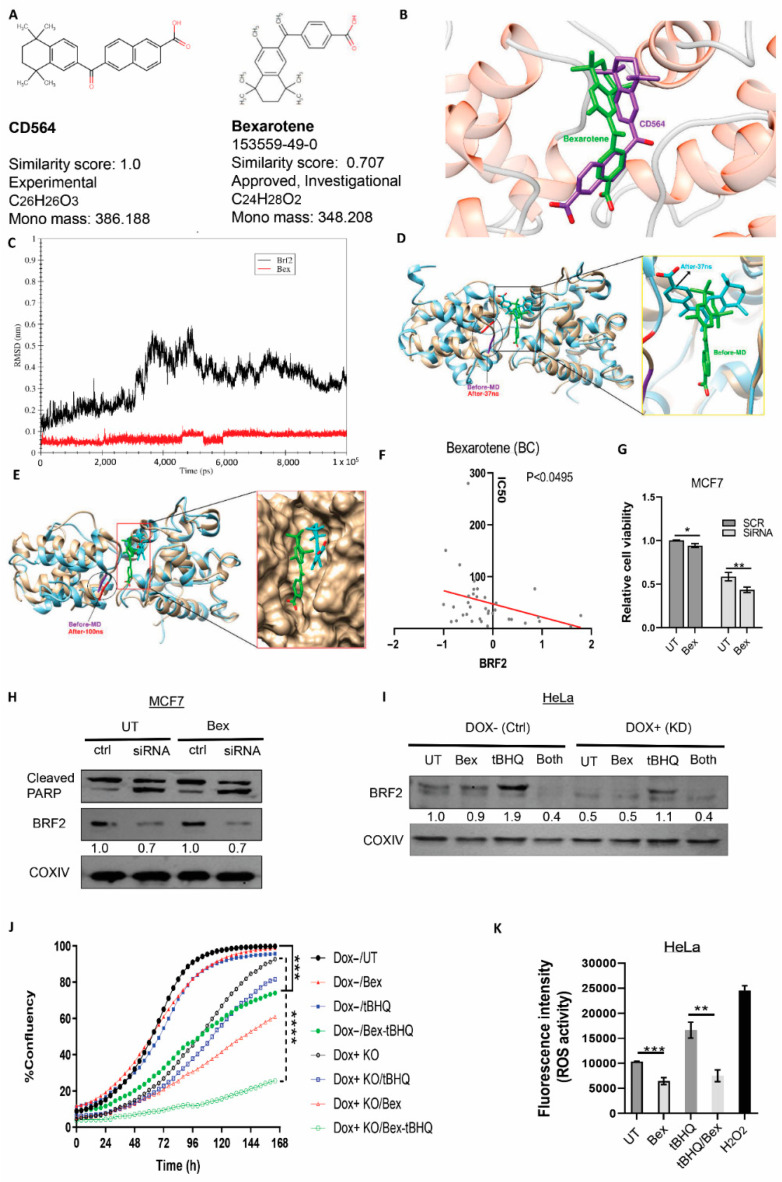
Effects of bexarotene (Bex) on BRF2. (**A**) Two-dimensional structure, chemical formula, and similarity score between CD564 (**left**) and Bex (**right**). (**B**) Docked BRF2-ligand complex for CD564 (purple) and Bex (green). (**C**) RMSD plot for BRF2-Bex during 100 ns of MD simulation. (**D**) Docking shows 2 conformations of the drug (before MD: green, and after 37 ns: cyan. (**E**) Docking shows 2 conformations of the drug (before MD: green, and after 100 ns: cyan. Note that the conformation of the drug has returned to its normal form. (**F**) Correlation between IC50 of Bex and BRF2 expression in breast cancer cell lines (threshold of expression is set between −2 and 2). (**G**) Relative cell viability comparison of untreated (UT) and drug-treated (Bex) cells between scramble (black) and siRNA-mediated knockdown of BRF2 (grey). (**H**,**I**) Immunoblotting showed the expression (H) of cleaved PARP and BRF2 in scrambled (Ctrl) and siRNA-mediated knockdown of BRF2 in the MCF7 cell line, (**I**) BRF2 expression in untreated (UT) cells, drug-treated (Bex) cells, tBHQ-induced oxidative stress (tBHQ), combination of Bex and tBHQ (Both) -treated cells, in CRISPER/inducible cas9-mediated knockdown of BRF2 in HeLa cells without doxycycline (Dox negative/Ctrl) and with 2 µg/mL doxycycline (Dox positive/Knockdown of BRF2). COX IV served as a loading control and the normalized quantification of BRF2 bands has been shown below each band. (**J**) Proliferation assay showing the percentage of confluency of HeLa cells without doxycycline (Dox-/control) and with 2 µg/mL doxycycline (Dox+/BRF2 knockdown) for each indicated condition. (**K**) ROS activity of HeLa cells for the indicated conditions (Untreated (UT) cells, bexarotene (Bex)-treated cells, tBHQ-treated cells, and both Bex + tBHQ treated cells. H_2_O_2_ served as a positive control) throughout the indicated time. For all calculations, data represent mean ± SD and data were measured in duplicate across 3 independent experiments. Student’s *t*-test was performed; * *p* < 0.05, ** *p* < 0.01, *** *p* < 0.001, **** *p* < 0.0001.

**Table 1 cancers-13-03778-t001:** List of antibodies used in this paper.

**Primary Antibodies**	
Polyclonal rabbit anti-COX IV (WB-1:1000)	Millennium Science Mulgrave, Vic, Australia
Monoclonal Mouse anti-yH2AX (WB, IF-1:1000)	Merck Millipore, Darmstadt, Germany
Polyclonal rabbit anti-BRF2 (WB-1:1000, IF-1:400)	Sapphire Bioscience, Redfern, NSW, Australia
Polyclonal rabbit anti-p53-s15 (WB-1:1000)	Sapphire Bioscience, Redfern, NSW Australia
Polyclonal rabbit anti-phospho KAP1 (WB-1:1000)	Bethyl laboratories, Inc. USA
Monoclonal Mouse anti-RAD51 (WB-1:1000)	GenTex Inc., USA
Polyclonal Rabbit anti-NRF2 (WB-1:1000)	Santa Cruz Biotechnology Dallas, Texas, USA
Monoclonal Mouse anti-NBS1 (WB-1:1000)	Becton Dickinson Biosciences, USA
Polyclonal Rabbit anti-MRE11 (WB-1:1000)	Becton Dickinson Biosciences, USA
Polyclonal Rabbit anti-H3 (WB-1:1000)	Abcam^®^, Cambridge, MA, USA
**Secondary Antibodies**	
Polyclonal anti-rabbit-antibody (IgG) Peroxidase-conjugate (from goat) (WB-1:3000)	Merck Millipore, Darmstadt, Germany
Polyclonal anti-mouse-antibody (IgG) Peroxidase-conjugate (from goat) (WB-1:3000)	Merck Millipore, Darmstadt, Germany

**Table 2 cancers-13-03778-t002:** Fifteen top-ranking docked compounds and their binding energies.

Target	Best Ligands	Binding Energy (kcal/mol)	Function
Protein: BRF2 PDB: 4ROC	Phthalocyanine	−10.5	An drug that is under investigation in clinical trial NCT00103246 for the treatment of patients with skin cancer, Bowen’s disease, actinic keratosis, and stage I/II mycosis fungoides.
	CD564	−10.1	Under investigation
	Alectinib	−9.9	A second-generation oral drug that selectively inhibits the activity of anaplastic lymphoma kinase (ALK) tyrosine kinase.
	Zk-806450	−9.8	Under investigation
	SR11254	−9.7	Under investigation
	BMS184394	−9.7	Under investigation
	AZD-5991	−9.7	An drug under investigation in clinical trial NCT03218683 for relapsed or refractory hematologic malignancies.
	Bemcentinib	−9.6	Bemcentinib has been investigated for the treatment of non-small cell lung cancer.
	Pranlukast	−9.6	A cysteinyl leukotriene receptor-1 antagonist which antagonises/reduces bronchospasm caused in asthmatics by an allergic reaction to accidentally or inadvertently encountered allergens.
	DB07456	−9.5	Under investigation
	TMC-647055	−9.5	TMC647055 has been used in trials studying the treatment of hepatitis C, chronic hepatitis C virus
	Aleplasinin	−9.5	Aleplasinin has been investigated in the treatment of Alzheimer’s disease.
	Orvepitant	−9.5	Orvepitant has been used in trials studying the treatment of depressive disorder and post-traumatic stress disorder (PTSD).
	Galicaftor	−9.4	A drug under investigation in clinical trial NCT03540524 with or without GLPG2737 in patients with cystic fibrosis.
	Baloxavir	−9.3	A drug under investigation in clinical trial NCT04327791 in combination therapy with baloxavir/oseltamavir 1 for patients with influenza.

**Table 3 cancers-13-03778-t003:** Calculation of binding free energy (ΔG) and components of ligand binding energy at the site of binding.

(kCal/mol)	∆G_vdw_	∆G_elec_	∆G_solv-polar_	∆G_solv-nonpol_	∆G_MMPBSA_
Native	−163.389	−427.767	197.664	−18.567	−412.059

## Data Availability

All the experimental data presented in this study are available in [article or Appendix A which can be found at: https://www.mdpi.com/2072-6694/13/15/3778/htm]. The datasets for all bioinformatic analysis from publicly available data sources are cited accordingly.

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
