# Peer review of "Targeting BRF2 in Cancer Using Repurposed Drugs"

_cancers, 2021, doi:10.3390/cancers13153778_

Round 1

Reviewer 1 Report

Good to see the authors addressed the second question and added relevant replies in the revised manuscript.    However, since authors cannot completely root out the possibility of BRF2's upregulation triggered by cisplatin-induced oxidative stress, authors may still want to acknowledge this possibility in writing.   Also, previous work using MM/PBSA/GBSA to evaluate the binding free energy between protein and ligands should be cited -  https://www.mdpi.com/1999-4923/13/7/927/htm https://www.thno.org/v08p0830.htm "

Author Response

Thank you for encouraging us to submit a revised version of the manuscript. We would like to thank the reviewers for their input and have read their comments with great interest. We believe we have satisfactorily addressed the reviewers’ comments. Detailed responses to reviewer’s comment are enclosed below.

We hope that the positive feedback from reviewers, together with the additional alterations done to address the reviewers’ comments, will allow you to accept this revised manuscript for publication in Special issue for Cancers.

Reviewer 1 is satisfied with the revision; we have incorporated his suggestion that we cannot exclude the possibility that BRF2 upregulation is triggered by cisplatin-induced oxidative stress. We have cited the relevant reference suggested by the reviewer for protein-ligand binding.

Reviewer 2 Report

Rashidieh et al. present a study demonstrating that BRF2 is a promising therapeutic target in cancer therapy. The authors further identified Bexarotene is a potential BRF2 inhibitor that reduces cellular ROS level and consequent BRF2 expression in cancer cells. Overall, the study showed good data quality, and the manuscript is prepared in a proper format of Cancers. However, the author did not provide the data demonstrating differential role of BRF2 between normal cells and cancer cells, especially those with BRAC1/2 dysfunction. The comments for the authors are listed as follows:

  1. On page 6, line 269, did the authors mean ATCC instead of ATTC?
  2. On page 8, line 360, H202 should be corrected.
  3. On page 9, line 394, the authors mentioned that BRF2 amplification was correlated with BRCA1/2 loss of function. Please confirm this hypothesis by comparing BRF2 expression in MCF10A, BRCA1/2 wild-type and mutant breast cancer cell lines.
  4. In figure 1, gamma-IR was able to induce BRF2 and DNA-damage response in normal cells and breast cancer cells. The data is not convincing that BRF2 is an ideal target in cancer therapy. The data did not demonstrate the differential effects of treatments (radiation, tBHQ, cisplatin) on BRF2 and DNA-damage responses between normal and cancer cells. How is the effect of these treatments on the viability of normal and cancer cells?
  5. On page 12, line 466, the authors concluded that there is a link between BRF2 upregulation and DNA damage induction, without the confounding effect of oxidative stress. Since NRF2 is not the only marker of ROS, the authors need to check ROS levels under these treatments. Can ROS scavenger rescue irradiation- or tBHQ/Cisplatin-induced DNA damage responses?
  6. The basal level of gamma-H2AX was not consistent in Fig. 1A and Fig. 1G.
  7. In Fig. 7F, the data is not significant since the P value is higher than 0.05.
  8. Again, if BRF2 is a good therapeutic target, the authors need to compare the effect of Bex and BRF2 knockdown on the viability of normal and cancer cells, especially those with BRCA1/2 mutation.
  9. On page 21, line 678, it is confusing that the authors use “knockdown”, instead of knockout in the CRISPR/iCas9 system. If BRF2 could not be knockout at the genomic level, how come the authors did not use siRNA- or shRNA-mediated knockdown cells to perform the experiments in Fig. 7I to 7K?
  10. On page 21, line 675, please correct the type “this this”.
  11. On page 21, line 680, Doxycycline induced (knockdown of BRF2) should be corrected, in case the readers might be confused.
  12. In Fig. 7F, there is a negative correlation between the IC50 values of Bexarotene and BRF2 levels in cancer cells. The data is not consistent with that in Fig. 7J, since BRF2-KD cells are more sensitive to Bex treatment.

Author Response

Thank you for encouraging us to submit a revised version of the manuscript. We would like to thank the reviewers for their input and have read their comments with great interest. We believe we have satisfactorily addressed the reviewers’ comments. Detailed responses to reviewer’s comment are enclosed below.

We hope that the positive feedback from reviewers, together with the additional alterations done to address the reviewers’ comments, will allow you to accept this revised manuscript for publication in Special issue for Cancers.

Reviewer 2  

Rashidieh et al. present a study demonstrating that BRF2 is a promising therapeutic target in cancer therapy. The authors further identified Bexarotene is a potential BRF2 inhibitor that reduces cellular ROS level and consequent BRF2 expression in cancer cells. Overall, the study showed good data quality, and the manuscript is prepared in a proper format of Cancers. However, the author did not provide the data demonstrating differential role of BRF2 between normal cells and cancer cells, especially those with BRAC1/2 dysfunction. The comments for the authors are listed as follows:

 We would like to thanks the reviewer for their comments and corrections.

  1. On page 6, line 269, did the authors mean ATCC instead of ATTC?

Yes, we have changed the typographical error.

  1. On page 8, line 360, H202 should be corrected.

Has been corrected.

  1. On page 9, line 394, the authors mentioned that BRF2 amplification was correlated with BRCA1/2 loss of function. Please confirm this hypothesis by comparing BRF2 expression in MCF10A, BRCA1/2 wild-type and mutant breast cancer cell lines.

We have previously reported that BRF2 is expressed at very low levels in near normal mammary epithelial cell lines (MCF10A and BRE80TERT), and is highly expressed in number of breast cancer lines and its expression was lower in HCC1937 and MDA MB 436 with one allele mutated and other affected by loss of heterozygosity (Figure 6a from our published paper by Srihari S, Mol Biosyst 2016, 12:963).

  1. In figure 1, gamma-IR was able to induce BRF2 and DNA-damage response in normal cells and breast cancer cells. The data is not convincing that BRF2 is an ideal target in cancer therapy. The data did not demonstrate the differential effects of treatments (radiation, tBHQ, cisplatin) on BRF2 and DNA-damage responses between normal and cancer cells. How is the effect of these treatments on the viability of normal and cancer cells?

In our previous paper, we have provided evidence that cancer cell lines with high BRF2 expression are dependent on it for survival ((Srihari S Mol Biosyst 2016, 12:963; Fig 6C) compared to breast cancer lines and normal mammary epithelial lines with low levels of BRF2, suggesting BRF2 as an ideal therapeutic target in breast cancer.  BRF2 is induced after treatment with radiation, tBHQ and cisplatin both in normal cells and cancer cells. There is a wealth of literature that indicates that cell fate after DNA damaging treatment is different in normal compared to cancer cells. Cancer cells tend to be more susceptible to DNA-damage induced cell death, due to defects in cell cycle checkpoints and DNA repair, compared to normal cells. Furthermore, growth inhibition of near normal mammary epithelial cells (MCF10A), HCC38, MCF7, MDAMB436 has been compared in response to cisplatin (Larson et al Scientific Report (2020) 10: 5798.  GR50 value (50% growth inhibition) was higher for MCF10A (29±4 µM) compared to breast cancer lines with Wt BRCA (MCF7=15 ± 10 µM and HCC 38=21± 7 µM) and BRCA1-mutant line MDA-MB 436 with homology-directed repair deficiency was most sensitive (9.4±1.7 µM).

We also present data for the reviewer that shows that near normal cells are less susceptible to apoptosis (assessed as Sub-G1 peak by flow cytometry) induction 48 and 72 hours after BRF2-depletion compared to a breast cancer line.

  1. On page 12, line 466, the authors concluded that there is a link between BRF2 upregulation and DNA damage induction, without the confounding effect of oxidative stress. Since NRF2 is not the only marker of ROS, the authors need to check ROS levels under these treatments. Can ROS scavenger rescue irradiation- or tBHQ/Cisplatin-induced DNA damage responses?

In our experimental condition, we have measured ROS generation after cisplatin treatment and were not able to demonstrate significant changes in ROS levels this has been stated in revised version lines 801-804 . Additionally, we have pre-treated MCF10A, MDAMB436 with 2mM NAC prior to IR exposure and have still seen IR-induced increase of BRF2, P-KAP1 and g-H2AX used a DNA damage marker (the figure below included for reviewer only; as the experiment was only been done once in triplicate during  10 days of revision time that encompassed lockdown of 5 days due to community transmission of COVID).

  1. The basal level of gamma-H2AX was not consistent in Fig. 1A and Fig. 1G.

We agree with reviewer, the Fig1G was over contrasted and we used the lower exposure to adjust the contrast.

  1. In Fig. 7F, the data is not significant since the P value is higher than 0.05.

Thank you for raising this typo issue, the p-value is less than (<) 0.0495, which is significant and the figure has been corrected.

  1. Again, if BRF2 is a good therapeutic target, the authors need to compare the effect of Bex and BRF2 knockdown on the viability of normal and cancer cells, especially those with BRCA1/2 mutation.

We have previously reported that breast cancer lines with high BRF2 expression are dependent on it for cellular viability (Srihari S Mol Biosyst 2016, 12:963). The viability of MCF10A is less impacted by depletion of BRF2 compared to other BRF2 high-expressing lines (fig 4C, from published paper). In this study, we have identified Bex as an interesting hit compound for the development of BRF2-specific inhibitors. We show that Bex does not impact BRF2 expression under basal condition, but it did dramatically reduce tBHQ-induced increase in BRF2 levels which was in-part mediated by ROS-scavenging activity of Bex. Bex is a third-generation synthetic retinoid that selectively activates retinoid X receptors which are known to have antioxidant activity. Further studies are, therefore, required to improve the selectivity of Bexarotene for BRF2 as the anti-proliferative effect of Bex alone or tBHQ plus Bex treatment observed in our study appears to be non-selective for BRF2 expression.

This point has been discussed in the discussion and conclusion sections.

  1. On page 21, line 678, it is confusing that the authors use “knockdown”, instead of knockout in the CRISPR/iCas9 system. If BRF2 could not be knockout at the genomic level, how come the authors did not use siRNA- or shRNA-mediated knockdown cells to perform the experiments in Fig. 7I to 7K?

We have used constitutive shRNA or siRNA to knockdown BRF2 for short-term experiments.  We have used an inducible Crisper-Cas9 system to knockdown BRF2 for uniform temporal control and efficient disruption for long-term maintenance of cell line without the knockout and to be able to induce BRF2-Knockout when required.

  1. On page 21, line 675, please correct the type “this this”.

Has been corrected

  1. On page 21, line 680, Doxycycline induced (knockdown of BRF2) should be corrected, in case the readers might be confused.

Has been corrected

  1. In Fig. 7F, there is a negative correlation between the IC50 values of Bexarotene and BRF2 levels in cancer cells. The data is not consistent with that in Fig. 7J, since BRF2-KD cells are more sensitive to Bex treatment.

In large-scale breast cancer cell line dataset, we did see a trend of negative correlation between IC50 of Bex and BRF2 levels ( -2< TCGA expression < 2   )in cancer cells, but it did not hold up in testing of individual cell lines as BRF2-depleted cells were even more sensitive to Bex treatment suggesting that Bex is not that selective for BRF2. Further work is required to improve its selectivity for BRF2. We have clarified this in sections of the manuscript on lines 824-826.

Round 2

Reviewer 2 Report

Thanks for the author's responses and I have no more question.

This manuscript is a resubmission of an earlier submission. The following is a list of the peer review reports and author responses from that submission.

Round 1

Reviewer 1 Report

  1. The whole manuscript is not well organized; a flow of work is lacking.
  2. Simple summary: In this section, only virtual screening part is mentioned.
  3. Introduction: line number 62, “Pol II I complex” should be Pol III complex.
  4. Line number 86-91, “detailed analysis of molecular events that disrupt BRF2-TBP complex.” Here what kind of molecular event analysis is performed?
  5. Line number 95, what is the meaning of 18 measures?
  6. Check the references in Materials and Methods section, remove all PMIDs and arrange according the journal’s format.
  7. Materials and Methods, 2.1,2.2 and 2.5 should be concise.
  8. Line number 176-177, “8000 molecular docking simulations” explain it?
  9. Result: Line number 356-359, it should be moved to discussion section.
  10. Representation of Figure 5 A and B should be similar as others (5C,D…). Ligplot (2D interaction) is not much clear.
  11. In order to explore more details, the best-ranked ligands (Pathalocyanine and CD564) should be presented in a separate figure.
  12. In the manuscript there are some simple spelling, comma and spacing problems, review it carefully.

Reviewer 2 Report

BRF2, as an essential component for the RNA Polymerase III and a transcription factor known for oxidative stress regulation, has been related to several cancers and thus a potential drug target. In this manuscript, the authors performed Pan-Cancer data analysis to show BRF2 expression tends to be upregulated in several cancers. The authors proposed a novel function of BRF2, based on both correlation of mutation events and cell assays, in DNA damage response reflected on the post-transcriptional but not transcriptional level, which is independent of the oxidative stress pathway. The authors then performed virtual drug screening using docking on more than 3500 FDA-approved and drugs under clinical trials on an MD-relaxed BRF2 structure and identified CD564 as a high-affinity drug targeting on the BRF2-TBP binding interface suggested by MD simulations and MM/PBSA-based binding free energy calculation. The current manuscript would advance the understanding of BRF2's role and clinical application for cancer, the authors could further address the following comments before its publication -

Major concerns:

  1. The manuscript is harmed by wrongly placed figures, wrongly positioned labels, and typos. Just list a few: the figures 2D and 2E should be reversed; the labels are put in the wrong column for several gels in figure 2. In table 2, the unit is in kcal/mol, however, in the manuscript, while the numbers are the same, the authors stated that the energy is in kJ/mol; the compressed file for the supplementary file is broken. Please also see minor issues below and address all issues similar as mentioed throughout the manuscript.

  1. BRF2 is known as the regulator of oxidative stress that can cause DNA damage. One major claim of this manuscript is that BRF2 could respond to DNA damage that is independent of the pathway responded to oxidative stress. The authors want to claim BRF2 upreglation can be triggered by DNA damage alone in the absence of oxidative stress. They showed in Figure 2F, where Nrf2, an oxidative stress biomarker, is not upregulated over 6 hrs given the DNA damage-introducing agent, cisplatin. However, it has been shown in a couple of papers that cisplatin can cause oxidative stress (Wangie Et al., 2018, Sci. Rep.; Martins et al., 2008, J Appl Toxicol). On the other hand, the figure panels 2A-2E could be explained as that the introduced oxidative stress could upregulate both oxidative stress-related (including BRF2) and DNA damage-related biomarkers. Figure 2G to 2J could also be explained as the presence of BRF2 helps reduce the oxidative stress and therefore cause less DNA damage. The proposed argument seems weak based on currently presented results.

  1. The authors have chosen the second-ranked drug from the docking results and used normal MD, umbrella sampling, and MM/PBSA to show its high affinity to the binding pocket. It was stated “CD564 … fits at the pocket of the site better than the large aromatic macrocyclic compound, Phthal-ocyanine”. It is hard to picture what criteria is used here to evaluate fitting quality. As shown in Figure 1F, BRF2 could have significant conformational change during the MD simulations in its free state, BRF2 could possibly adapt to a proper conformation with the top-ranked drug and resulting in a high affinity of the binding. Given the success of AutoDock Vina in prioritizing the high-affinity drug, of -412.059 kJ/mol based on MM/PBSA, in the second place, the authors should also consider running some MD simulation and binding free energy (say, MM/PBSA) estimation on more top-ranked drugs including Phthalocyanine, the number 1 drug from docking (and prioritize the drugs based on MM/PBSA but not the poor docking forcefield).

Minor concerns:

  1. line 62, "Pol II I" should be "Pol III".
  2. line 141, "these twelve pathways" were not further specified.
  3. line 171, "8.770" should be "8,770".
  4. From line 177 to line 180, "which included ten docking runs." could be considerably modified as "which generated ten docking poses." The searching algorithm implemented in AutoDock Vina is different from AutoDock. The authors could consider removing the following sentence without impairing the clarity of the presented method: " Generated by the Lamarckian Genetic Algorithm, 10 binding poses representative of the cluster centroids of all conformations are selected by the AutoDock Vina program."
  5. Following the above commend, the authors could consider reporting the "exhaustiveness" value of the parameters used in running AutoDock Vina.
  6. line 201, a redundant "the" is here.
  7. line 248 and 249, the authors should give the citations where the parameters are derived.
  8. line 251, the authors may want to confirm they used "the average structure" or "the structure closest to the average structure".
  9. line 268, how did the authors choose the 27 configurations out of 500?
  10. line 268, "DG" should be "ΔG".
  11. Figure 3, no explanation for the color scheme in the caption
  12. Figure 3C, the color scheme could consider matching those colored in panels A, B, and D.
  13. line 491, cannot get from the context the meaning of "the number of deviations" and how it can be used to assess the stability of the conformation.
  14. line 498, "domain 2" is not defined throughout the manuscript.
  15. Figure 4D, the figure plot only one PC, instead of the first two PCs as described in the main text. Also, there is no description in the caption specifying which PC is plotted.
  16. line 523, authors could add the description of getting RMSF profile of protein residues from PC modes in the "2.7. Principal Component Analysis (PCA)".
  17. line 530, Figure 6 does not contain the figure comparing the free and TBP-DNA-complexed BRF2.
  18. line 533, "FEL" should give its full name for being mentioned for the first time in the manuscript.
  19. line 548, the authors do not describe how they select the optimized structure of BRF2 from the MD trajectory.
  20. Table 1, the format of column label "BindingEn" should be adjusted.
  21. line 580, "∆Gsolv-polar" might be "∆G vdw"
  22. line 585, "DG" should be "∆G".
  23. line 578, the authors should give an exact number of ∆G for the electrostatic interaction contributed by Arg81.
  24. line 587, "Figure XA".
  25. line 593, "Figure XF"
  26. Figure 6B, the pocket contains two ligands without proper explanation in the captions.
  27. line 606, "100 ns MD simulation" instead of “100 MD simulation”

Reviewer 3 Report

This is an interesting manuscript that provides an explanation for the upregulation of BRF2 found in certain cancers. The authors provide experimental confirmation of the link between BRF2 and DNA damage response signaling. Upon finding changes in the conformation of the TBP-DNA complex upon BRF2 binding, the authors performed extensive MD and docking simulations to characterize these interactions. They then embarked on a virtual screening experiment utilizing the information gleaned from their simulations to identify potential inhibitors, then showed by docking how their lead molecule, CD564, is an excellent fit for the pocket, and could be a potential inhibitor that functions by inhibiting BRF2.

The authors have rigorously performed computational studies, and extensively analyzed and identified key drivers of molecular interactions with inhibitors in the binding pocket. Such studies are no doubt essential, and very useful, for medicinal chemistry and drug development. However, the authors missed the key step of experimentally confirming their findings - do the top molecules really inhibit BRF2 binding? Does binding inhibition lead to cell death? Is the inhibition selective? Does the scoring system reflect BRF2 inhibitory potential i.e, do the low scoring molecules have lower affinity and lower IC50 values?

Like the authors correctly identified, their studies need criticial experimental validation. Such studies would have completed the story and made for a more compelling story. As written, the manuscript may be more suitable for a computational chemistry focused journal.